# Sparse genetic tracing reveals regionally specific functional organization of mammalian nociceptors

William Olson[1], Ishmail Abdus-Saboor[1†], Lian Cui[1†], Justin Burdge[1], Tobias Raabe[2], Minghong Ma[1], Wenqin Luo[1]*

[1]Department of Neuroscience, Perelman School of Medicine, University of Pennsylvania, Philadelphia, United States; [2]Department of Genetics, Perelman School of Medicine, University of Pennsylvania, Philadelphia, United States

**Abstract** The human distal limbs have a high spatial acuity for noxious stimuli but a low density of pain-sensing neurites. To elucidate mechanisms underlying regional differences in processing nociception, we sparsely traced non-peptidergic nociceptors across the body using a newly generated $Mrgprd^{CreERT2}$ mouse line. We found that mouse plantar paw skin is also innervated by a low density of Mrgprd+ nociceptors, while individual arbors in different locations are comparable in size. Surprisingly, the central arbors of plantar paw and trunk innervating nociceptors have distinct morphologies in the spinal cord. This regional difference is well correlated with a heightened signal transmission for plantar paw circuits, as revealed by both spinal cord slice recordings and behavior assays. Taken together, our results elucidate a novel somatotopic functional organization of the mammalian pain system and suggest that regional central arbor structure could facilitate the "enlarged representation" of plantar paw regions in the CNS.

DOI: https://doi.org/10.7554/eLife.29507.001

*For correspondence: luow@
pennmedicine.upenn.edu

†These authors contributed equally to this work

Competing interests: The authors declare that no competing interests exist.

## Introduction

The skin mediates physical contact with environmental mechanical, thermal, and chemical stimuli. As an animal moves through the world, certain parts of the skin are likely sites of 'first contact' with these stimuli (for instance, the distal limbs, face/whiskers, and tail for quadrupedal mammals like mice). Therefore, these somatosensory regions require heightened sensitivity to fulfil behaviorally relevant functions, such as environment exploration.

Touch and pain are the two most important types of somatosensation for this functional purpose: touch allows for feature detection while pain prevents serious tissue damage. Indeed, classic work has defined important regional specialization of the nervous system for tactile sensation in these areas. Two mechanisms in the peripheral organization of the discriminative touch system facilitate high spatial acuity sensation in the primate distal limbs and mouse whisker pad. These are the *high innervation density* and *smaller receptive field sizes* of the primary light touch neurons, the Aβ mechanoreceptors, in these regions (*Johansson and Vallbo, 1979*; *Johansson and Vallbo, 1980*; *Brown and Koerber, 1978*; *Weinstein, 1968*; *Brown et al., 2004*; *Paré et al., 2002*; *Rice et al., 1993*). In contrast, the question of whether regional specialization exists in the mammalian pain system has remained elusive until recently. Upon stimulation using nociceptive-specific laser beams, human subjects show a heightened spatial acuity in the distal limbs (especially the fingertips) for pain stimuli, much like they do for touch stimuli (*Mancini et al., 2012*, *2013*, *2014*). This suggests that this region is also a 'pain fovea' for humans. However, human fingertip skin has a relatively *low density of pain-sensing neurites* (*Mancini et al., 2013*). While this suggests that region-specific

organization likely exists in pain circuits downstream of the periphery (i.e. central nervous system), currently the exact underlying neural mechanisms are unclear.

Noxious stimuli are detected by primary sensory neurons called nociceptors, which have cell bodies in the dorsal root ganglia (DRG) or trigeminal ganglia (TG) and axons that bifurcate into peripheral and central branches. The peripheral projection of a nociceptor usually terminates as a free nerve arbor in the skin or deeper tissues, while the central projection terminates in an arbor in the dorsal horn (DH) of the spinal cord or caudal medulla. Though previous work has mapped the peripheral receptive fields or traced the central terminals of mammalian nociceptors (*Schmidt et al., 1997*; *Schmidt et al., 2002*; *Bessou and Perl, 1969*; *Beitel and Dubner, 1976*; *Treede et al., 1990*; *Lynn and Carpenter, 1982*; *Lynn and Shakhanbeh, 1988*; *Sugiura et al., 1986*; *Sugiura et al., 1989*; *Szentágothai, 1964*; *Réthelyi, 1977*), these studies have not established a model for the somatotopic functional organization of the mammalian pain system due to the limited number of neurons traced from restricted skin regions.

We therefore sought to reveal the region-specific organization of mammalian nociceptors across the entire body. We generated a novel *Mrgprd*[CreERT2] mouse line to perform systematic sparse genetic tracing of a population of non-peptidergic nociceptors that mediate mechanical pain and beta-alanine (B-AL) triggered itch (*Cavanaugh et al., 2009*; *Liu et al., 2012*). We chose these neurons because they are the most abundant type of cutaneous polymodal high threshold C fibers (nociceptors), and they likely correspond to the main type of free nerve terminals stained with anti-PGP9.5 antibody in previous human skin biopsy data (*Mancini et al., 2013*; *Zylka et al., 2005*). Indeed, like the human skin biopsy results, we found that Mrgprd[+] neurites have a comparatively low density in plantar paw compared to trunk skin. Retrograde tracing experiments further show that the number of Mrgprd[+] neurons innervating per unit area of plantar paw glabrous skin is lower compared to those innervating upper hind limb hairy skin. In addition, and in contrast to the Aß mechanoreceptors, sparse genetic tracing revealed that the arbor field sizes of individual nociceptors are comparable between different skin regions. Strikingly, plantar paw and trunk innervating nociceptors display distinct morphologies in their central terminals. Moreover, using *Mrgprd*[CreERT2]; *Rosa*[ChR2-EYFP] mice, we specifically activated these nociceptors using blue laser light during in vitro spinal cord slice recordings and during behavior assays. We found that, while almost all layer II DH neurons in all locations receive direct Mrgprd[+] afferent input, the optical threshold required to induce postsynaptic responses is much lower in plantar paw regions. This was paralleled by a decrease in the light intensity threshold required to elicit a withdrawal response in paw, compared to upper thigh, skin stimulation. Collectively, we have identified a previously unappreciated somatotopic difference in the central terminals of mammalian nociceptors. Our anatomical, physiological, and behavior data suggest that region-specific central arbor structure could be one important mechanism to magnify the representation of plantar paw nociceptors in the DH to facilitate region-specific pain processing.

## Results

### Generation and specificity of *Mrgprd*[CreERT2] mice

Given that previous descriptions of nociceptor structure have not allowed for systematic comparisons between body regions, we sought to use sparse genetic labeling to trace single nociceptor morphologies across the entire somatotopic map. We generated a mouse line in which a tamoxifen-inducible Cre (CreERT2) cassette is knocked into the coding region of Mas-related gene product receptor D (*Mrgprd*) (*Figure 1A*, *Figure 1—figure supplement 1*). Consistent with the previous finding that *Mrgprd* is expressed more broadly in early development than in adulthood (*Liu et al., 2008*), early embryonic (E16.5-E17.5) tamoxifen treatment of *Mrgprd*[CreERT2] mice labels *Mrgprd* expressing neurons along with non-peptidergic neurons expressing other *Mrgpr* genes, such as *Mrgpra3* and *Mrgprb4* (*Figure 1—figure supplement 2*). In contrast, when we crossed *Mrgprd*[CreERT2] mice with a Cre-dependent *Rosa*[ChR2-EYFP] line and provided postnatal (P10-P17) tamoxifen treatment (*Figure 1B*), Mrgprd[+] non-peptidergic nociceptors were specifically labeled. We examined these treated mice at four postnatal weeks or older (>4 pw), a time point at which Mrgprd[+] non-peptidergic nociceptors have completely segregated from other Mrgpr[+] DRG neurons (*Liu et al., 2008*). We found that ChR2-EYFP[+] DRG neurons bind IB4 (a marker for non-peptidergic

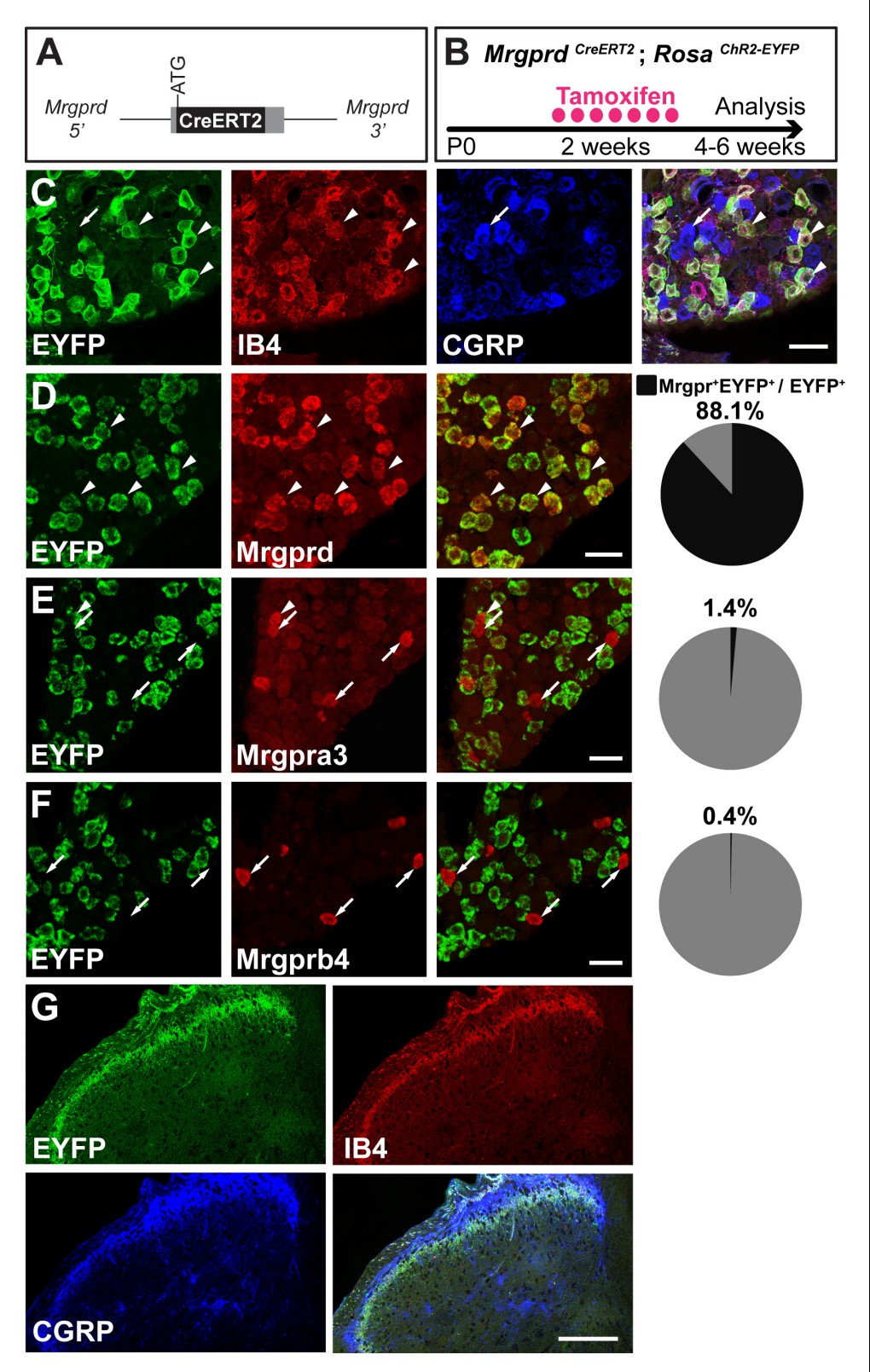

**Figure 1.** *Mrgprd^{CreERT2}* mice can mediate recombination specifically in adult Mrgprd⁺ non-peptidergic nociceptors. (A) Knock-in *Mrgprd^{CreERT2}* allele. (B) Illustration showing tamoxifen treatment scheme, 0.5 mg tamoxifen/day, P10-P17 treatment of *Mrgprd^{CreERT2}; Rosa^{ChR2-EYFP}* mice. (C) Triple staining of DRG section showing EYFP overlaps with IB4 but not CGRP. (D–F) Double fluorescent in situ DRG sections showing EYFP in *Mrgprd* (D) but not *Mrgpra3* (E) or *Mrgprb4* (F) cells. Pie charts show overlap quantification (% of EYFP⁺ cells that co-express Mrgpr, *n* = 3 animals). (G) DH section showing
*Figure 1 continued on next page*

*Figure 1 continued*

EYFP$^+$ terminal overlap with IB4 but not CGRP. Arrowheads show overlapping cells, arrows show non-overlapping cells. Scale bars = 50 μm (**C–F**), 100 μm (**G**).

DOI: https://doi.org/10.7554/eLife.29507.002

The following figure supplements are available for figure 1:

**Figure supplement 1.** Generation of *Mrgprd$^{CreERT2}$* knock-in mouse line.

DOI: https://doi.org/10.7554/eLife.29507.003

**Figure supplement 2.** Prenatal tamoxifen treatment labels Mrgprd$^+$ along with Mrgpra3/b4$^+$non-peptidergic DRG neurons.

DOI: https://doi.org/10.7554/eLife.29507.004

DRG neurons) but do not express CGRP (a marker for peptidergic DRG neurons) (*Zylka et al., 2005*) (*Figure 1C*), and ChR2-EYFP$^+$ DH terminals similarly overlap with IB4 but not CGRP (*Figure 1G*). Double in situ hybridization demonstrated that this strategy efficiently labels DRG neurons expressing *Mrgprd* (88.1 ± 1% of ChR2-EYFP$^+$ neurons, $n$ = 3 animals) but not those expressing *Mrgpra3* (1.4 ± 0.1%) or *Mrgprb4* (0.4 ± 0.3%) (*Figure 1D–F*). Almost all *Mrgprd* expressing neurons were labeled with ChR2-EYFP (92.9 ± 4.6% of *Mrgprd$^+$* neurons) by this treatment. Therefore, this newly generated inducible *Mrgprd$^{CreERT2}$* line allows for the specific and efficient targeting of adult Mrgprd$^+$ nociceptors.

We then sought to trace individual Mrgprd$^+$ non-peptidergic nociceptors using sparse genetic labeling. When crossed with a Cre-dependent alkaline phosphatase reporter line (*Rosa$^{iAP}$*), we found that sparse recombination occurs in the absence of tamoxifen treatment (*Figure 2A and B*). This background recombination labels 3–11 neurons/DRG (5.2 ± 1.6 neurons/DRG, $n$ = 47 DRGs from three animals) in 3–4 pw animals (*Figure 2C*), which represents <1% of the total Mrgprd$^+$ nociceptor population (*Zylka et al., 2005*; *Wright et al., 1997*; *Molliver et al., 1997*). The sparsely labeled DRG neurons co-express non-peptidergic nociceptor markers peripherin, PAP (*Zylka et al., 2008*), and RET (*Figure 2D,F,H*), but do not express NF200 or CGRP (*Figure 2E,G*). To further determine the specificity of this sparse recombination, we used an *Mrgprd$^{EGFPf}$* knock-in line (*Zylka et al., 2005*), in which expression of EGFP mimics the dynamic expression of endogenous *Mrgprd*. We generated *Mrgprd$^{CreERT2/EGFPf}$*; *Rosa$^{iAP}$* triple mice and found that almost all AP$^+$ neurons co-express *Mrgprd$^{EGFPf}$* (93. 7 ± 2.3%, $n$ = 126 AP$^+$ neurons from three animals) (*Figure 2I*) in 3 to 4pw mice. This result indicates that, although *Mrgprd* is broadly expressed during early development, this background recombination occurred preferentially in adult Mrgprd$^+$ nociceptors.

## Genetic tracing of Mrgprd$^+$ skin terminals reveals a relatively comparable organization in the periphery

Mrgprd$^+$ neurons innervate both hairy and glabrous skin and are the most abundant class of cutaneous free nerve arbors (*Zylka et al., 2005*). To systematically compare the peripheral single-cell structure of mammalian pain neurons across the somatotopic map, we performed whole mount colorimetric AP staining using untreated 3–4 pw *Mrgprd$^{CreERT2}$*; *Rosa$^{iAP}$* skin.

We found that 98.4% (130/132 arbors, $n$ = 4 animals) of single-cell arbors have a 'bushy ending' morphology (*Figure 3A and B,D and E*, *Figure 3—figure supplement 1A*) (*Wu et al., 2012*), featuring thickened terminal structures in the epidermis. The distal ends of arbors in glabrous plantar paw skin have single, un-branched thickened neurites (*Figure 3A and B*), while arbors in the hairy skin feature both un-branched neurites as well as dense neurite clusters (*Figure 3D and E*). These dense clusters form circumferential-like endings that innervate the necks of hair follicles (red arrowheads in *Figure 3E and F*, *Figure 3—figure supplement 1*) (*Zylka et al., 2005*). Whole mount immunostaining of *Mrgprd$^{EGFPf}$* skin shows that all three types of hair follicles in mouse hairy skin (guard, awl/auchenne, and zigzag [*Li et al., 2011*]) are innervated by *Mrgprd$^{EGFPf}$* fibers (*Figure 3—figure supplement 1D–H*). A very small minority (1.6%) of arbors in the hairy skin have "free endings" (*Wu et al., 2012*) lacking these thickened structures (*Figure 3—figure supplement 1A–C*).

Mrgprd$^+$ non-peptidergic nociceptive field sizes range from 0.08 to 0.9 mm$^2$, with the smallest average field size found in the head skin between the ears and in the proximal limbs (*Figure 3G*, *Figure 3—source data 1*). Interestingly, non-peptidergic nociceptors innervating the distal limbs (plantar and dorsal paw skin) have average arbor sizes close the middle of this range, and distal limb and

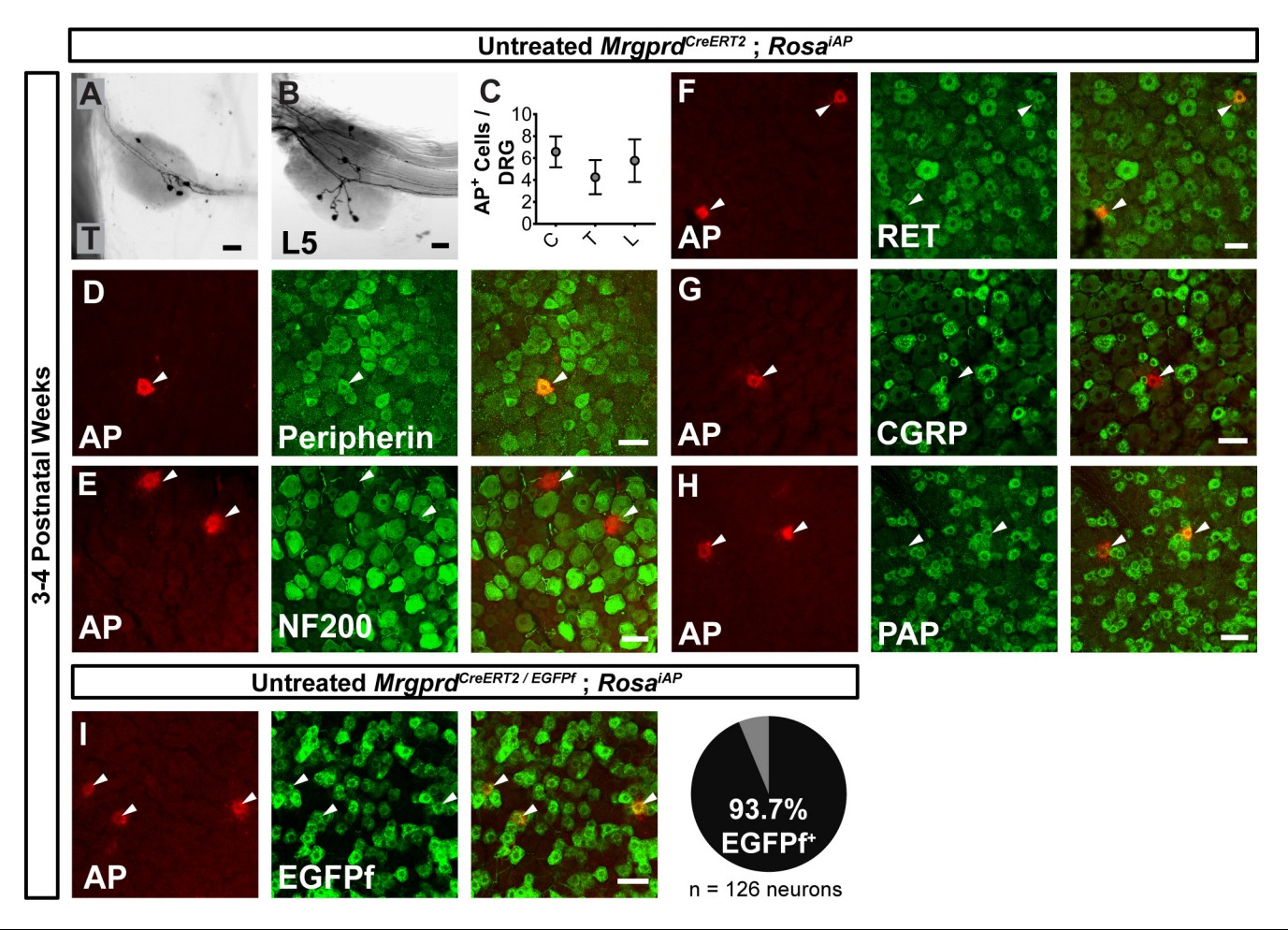

**Figure 2.** Sparse Mrgprd[+] nociceptor labeling in untreated 3–4 pw *Mrgprd*[CreERT2]; *Rosa*[iAP] mice. (A and B) Whole mount AP DRG staining of thoracic (A) and L5 (B) DRGs. (C) AP[+] cells / DRG for cervical (C), thoracic (T), and lumbar (L) DRGs, *n* = 47 DRGs from three animals. (D–H) Whole mount DRG immunostaining plus AP fluorescent staining. Sparse AP[+] cells express non-peptidergic nociceptor markers peripherin (D), RET (F) and PAP (H) but not large diameter neuron maker NF200 (E) or peptidergic marker CGRP (G). (I) Whole mount EGFPf immunostaining plus AP fluorescence staining of untreated *Mrgprd*[CreERT2/EGFPf]; *Rosa*[iAP] DRGs. AP[+] neurons are Mrgprd[+] nociceptors. Quantification of overlap (% of AP[+] cells that co-express *Mrgprd*[EGFPf], *n* = 126 neurons from three animals). Scale bars = 50 μm.

DOI: https://doi.org/10.7554/eLife.29507.005

trunk arbors are comparable in size (*Figure 3G and H*, *Figure 3—source data 1*). In addition, consistent with human skin (*Mancini et al., 2012*), whole-population labeling of Mrgprd[+] fibers using tamoxifen (0.5 mg at P11) reveals that the overall neurite density is similar or slightly lower in the paw glabrous skin compared to trunk hairy skin (*Figure 3C and F*). We further performed a retrograde DiI labeling experiment using *Mrgprd*[EGFPf] mice to compare the Mrgprd[+] neuron innervation densities (i.e. number of neurons innervating a unit area of skin) between paw and upper hind limb regions (*Figure 3—figure supplement 2*, *Figure 3—source data 2*). Consistent with the overall neurite density, we found that the neuron density is higher in the proximal hind limb compared to plantar paw (cells/mm$^2$ of skin: proximal hind limb = 97.6 ± 33.4, plantar hind paw = 21.0 ± 3.0, *n* = 7 injections for each, p=0.04, Student's t test) (*Figure 3I*). In short, in contrast to mammalian Aβ mechanoreceptors, plantar paw innervating mouse non-peptidergic nociceptors do not display higher density or smaller receptive field sizes compared to other regions.

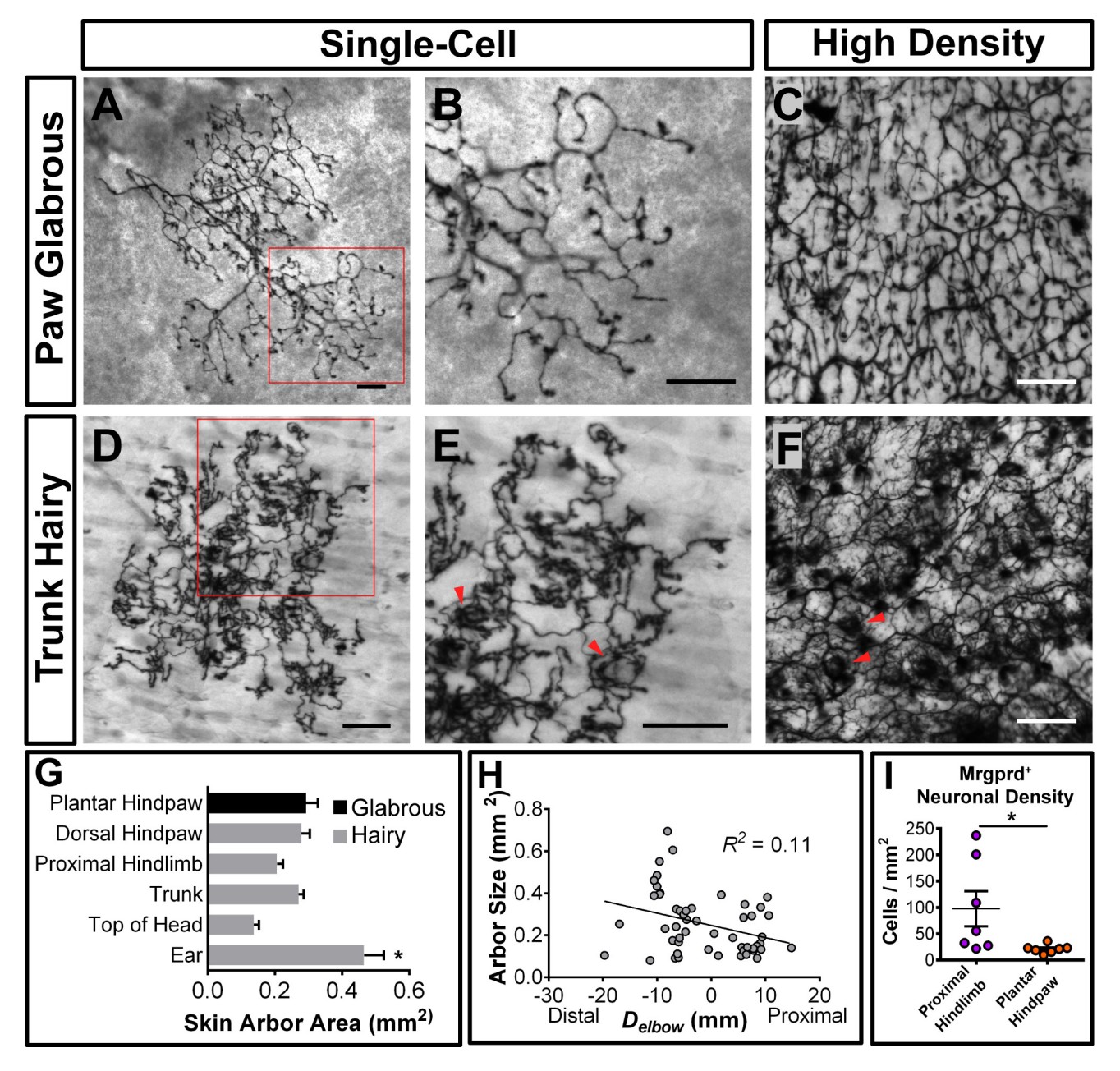

**Figure 3.** Peripheral organization of non-peptidergic nociceptors in 3–4 pw $Mrgprd^{CreERT2}$; $Rosa^{iAP}$ mice. (A, B, D, E) Sparse labeled non-peptidergic nociceptors show bushy-ending structure in the glabrous skin (**A–B**) and trunk hairy skin (**D–E**). B, E, high magnification images of regions boxed in A and D, respectively. (**C, F**) High-density labeled (0.5 mg tamoxifen at P11) glabrous and hairy skin. Overall neurite density is lower in glabrous compared to hairy skin. Red arrowheads in E and F mark neurite clumps that surround hair follicles. (**G**) Arbor areas in different skin regions. $n$ = 173 terminals from nine animals. *p<0.05 (one-way ANOVA with Tukey's multiple comparisons test). (**H**) Arbor areas in the hind limb skin vs. proximodistal distance ($D_{elbow}$) from the elbow (point 0, terminals distal to this edge are given negative $D_{elbow}$ values). $n$ = 52 arbors from four animals. No clear relationship between proximodistal location and size is evident (linear regression). (**I**) Mrgprd+ neuron density (number of retrogradely labeled DiI/$Mrgprd^{EGFPf}$ double positive neurons/area of DiI labeled skin, see **Figure 3—figure supplement 2**) is lower in plantar hind paw compared to proximal hind limb skin. *p<0.05 (Student's t test). Scale bars = 100 µm.

DOI: https://doi.org/10.7554/eLife.29507.006

The following source data and figure supplements are available for figure 3:

**Source data 1.** Summary of peripheral terminals of sparsely labeled Mrgprd+ non peptidergic nociceptors.

DOI: https://doi.org/10.7554/eLife.29507.009

*Figure 3 continued on next page*

*Figure 3 continued*

**Source data 2.** Retrograde DiI⁺labeling of nociceptors in *Mrgprd^EGFPf* mice.
DOI: https://doi.org/10.7554/eLife.29507.010
**Figure supplement 1.** Mrgprd⁺ fiber hairy skin innervation.
DOI: https://doi.org/10.7554/eLife.29507.007
**Figure supplement 2.** Retrograde DiI labeling of *Mrgprd^EGFPf* nociceptors.
DOI: https://doi.org/10.7554/eLife.29507.008

## Mrgprd⁺ nociceptors show regionally distinct organization in their central arbors

Since the peripheral organization of Mrgprd⁺ non-peptidergic nociceptors does not exhibit an obvious mechanism to facilitate heightened sensitivity in the plantar paw, we next used whole mount AP staining of untreated 3–4 pw *Mrgprd^CreERT2; Rosa^iAP* spinal cords to compare their central arbors between regions. Non-peptidergic nociceptor central branches enter the spinal cord through the dorsal root, travel rostrally or caudally for 0 to 3 segments, and then dive ventrally to establish arbors in layer II of the DH (*Table 1*) (*Zylka et al., 2005*). Most Mrgprd⁺ central branches do not bifurcate (65.8%, *n* = 234 neurons from 3 animals), and most also terminate within the segment of entry (72.6%) (*Table 1*). However, some (34.2%) bifurcate one or more times in the spinal cord, and some (27.3%) travel up to 3 segments from the point of entry (*Table 1*). For the central branches that bifurcate, most of their secondary/tertiary branches join other branches from the same neuron to co-form one axonal arbor, while some end with a second arbor or terminate in the spinal cord without growing an arbor (*Table 1*, *Figure 4—figure supplement 1A and B*). The majority (91.9%) of labeled nociceptors have only one arbor, but a few have 2 (6.8%), or 3 (0.9%) central arbors, and for a small number of Mrgprd + nociceptors (0.4%), we could not identify any arbor (*Table 1*).

Strikingly, Mrgprd⁺ non-peptidergic nociceptor central arbors display two different morphologies that can be distinguished by the ratio of their mediolateral width to their rostrocaudal height (W/H ratio) (*Figure 4H*). We defined arbors with W/H ratios > 0.2 as 'round' (*Figure 4A–C,H*, and *Figure 4—source data 1*) and arbors with W/H ratios < 0.2 as 'long and thin' (*Figure 4B–F,H*, and

**Table 1.** Summary of central innervation patterns of sparsely labeled Mrgprd⁺ non peptidergic nociceptors.
Data pooled from three 3pw animals.

|  | N | % of Total |
| --- | --- | --- |
| Total neurons | 234 |  |
| Segments traveled from point of entry |  |  |
| 0 | 170 | 72.6 |
| 1 | 54 | 23.1 |
| 2 | 9 | 3.8 |
| 3 | 1 | 0.4 |
| Direction traveled (for axons traveling 1–3 segments) |  |  |
| Caudal | 46 | 19.7 |
| Rostral | 18 | 7.7 |
| No central branch bifurcations | 154 | 65.8 |
| 1 central branch bifurcation | 72 | 30.8 |
| >1 central branch bifurcation | 8 | 3.4 |
| No central terminals | 1 | 0.4 |
| 1 central terminals | 215 | 91.9 |
| 2 central terminals | 16 | 6.8 |
| 3 central terminals | 2 | 0.9 |

DOI: https://doi.org/10.7554/eLife.29507.011

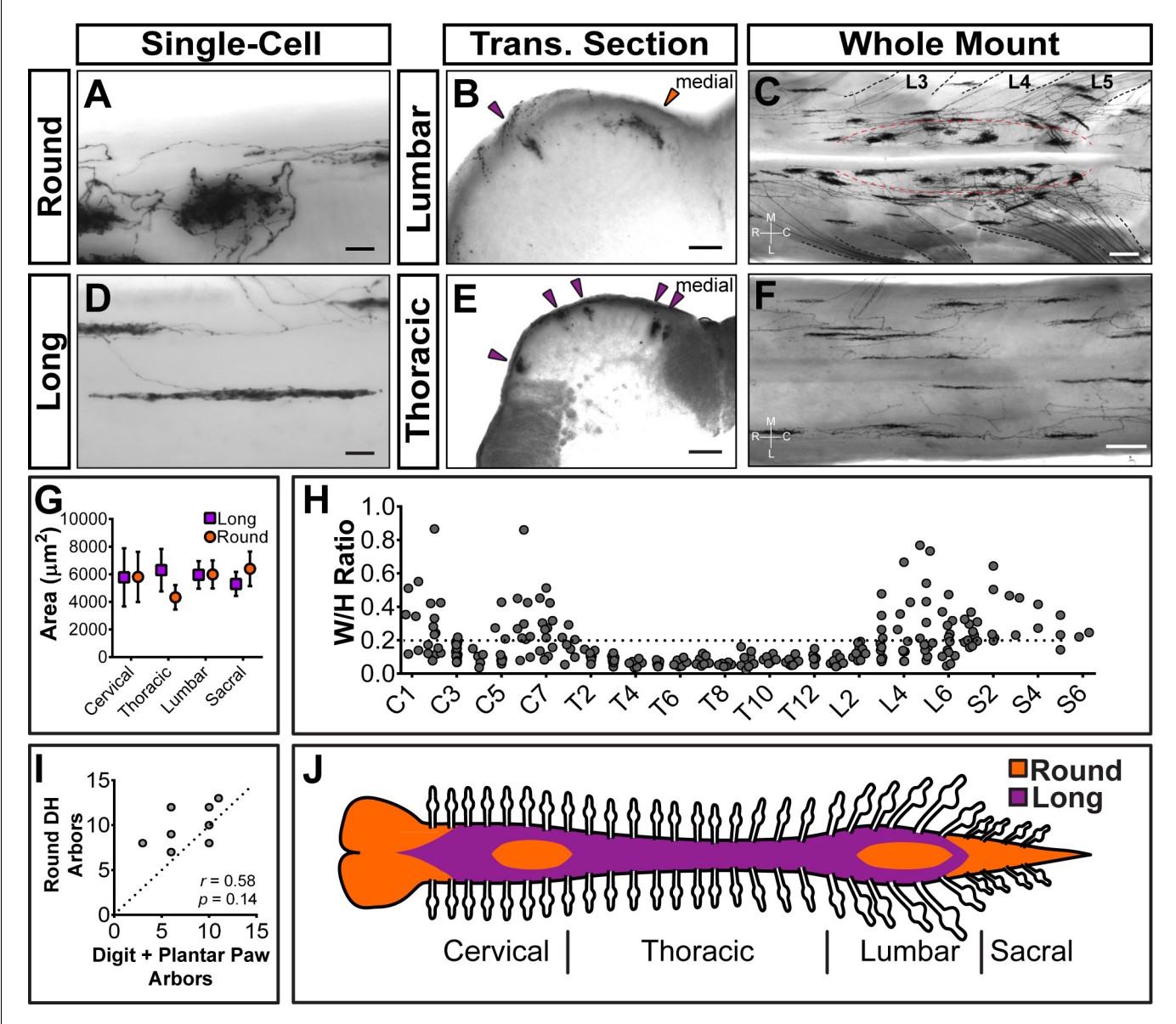

**Figure 4.** Sparsely labeled *Mrgprd*$^{CreERT2}$; *Rosa*$^{iAP}$ nociceptors have region-specific central arbor morphologies. (A–F) Round and long non-peptidergic central arbors seen in top-down whole mount (A, C, D, F) and transverse section (B and E) spinal cords. Round arbors are in the medial lumbar enlargement (B and C) while long arbors are in the lateral lumbar enlargement and thoracic spinal cord (B–F). Dorsal roots are outlined and labeled in C. Red dashed lines outline the round arbor zone. Orange arrowhead marks a round arbor, purple arrowheads mark long arbors. M, medial. L, lateral. R, rostral. C, caudal. (G) Round and long (defined by ratio in H) arbor areas are comparable for all regions. (H) Arbor Width/Height ratios by ganglion of origin. Round terminals: W/H > 0.2. *n* = 368 arbors from seven animals (I) Comparison of the number of labeled arbors in the hind limb digit and plantar paw skin with the number of ipsilateral round arbors in the dorsal horn. *n* = 4 animals, dotted line shows 1:1 relationship. *r, p* values from Spearman's rank correlation test. (J) Illustration showing the distribution of round (orange zone) and long (purple zone) arbors in the spinal cord. Scale bars = 50 μm (A and B, D and E), 250 μm (C and F).

DOI: https://doi.org/10.7554/eLife.29507.012

The following source data and figure supplement are available for figure 4:

**Source data 1.** Summary of non-peptidergic nociceptor central arbor height and width measurements.
DOI: https://doi.org/10.7554/eLife.29507.014
**Figure supplement 1.** Central arbors of sparsely labeled non-peptidergic neurons in *Mrgprd*$^{CreERT2}$; *Rosa*$^{iAP}$ mice.
DOI: https://doi.org/10.7554/eLife.29507.013

*Figure 4—source data 1*). Further, these morphological types are regionally segregated. Round arbors are found in DH regions known to represent the distal limbs (medial cervical and lumbar enlargements) as well as tail, anogenital (sacral DH), and head/face (descending trigeminal terminals in the upper cervical DH and medulla) skin (*Figure 4B and C,H,J,* and *Figure 4—figure supplement 1C*) (*Koerber and Brown, 1982*; *Molander and Grant, 1985*). Long arbors are instead located in regions corresponding to the proximal limbs (lateral cervical and lumbar enlargements) and trunk skin (thoracic DH) (*Figure 4B–F,H,J*) (*Koerber and Brown, 1982*; *Molander and Grant, 1985*; *Cervero and Connell, 1984*). While round and long arbors differ in morphology, they do not differ in area (*Figure 4G*).

In the cervical and lumbar enlargements (C3-C6, L3-L6), the medial DH contains a curved zone of round arbors that is encircled by laterally located long arbors (*Figure 4C*). Somatotopic mapping of the cat and rat DH indicates that this medial curved zone in the lumbar enlargement matches the representation of the plantar paw and digits, with the dorsal paw and proximal limb representations lying more laterally (*Koerber and Brown, 1982*; *Molander and Grant, 1985*; *Brown and Fuchs, 1975*; *Brown et al., 1991*; *Swett and Woolf, 1985*; *Takahashi et al., 2002*; *Takahashi et al., 2007*; *Woolf and Fitzgerald, 1986*). We compared the number of labeled peripheral arbors in the toe and plantar paw skin with the number of round DH terminals in the corresponding half of the lumbar enlargement. This showed a very close correlation (*Figure 4I*), supporting that the round central arbors of the lumbar enlargement correspond to plantar paw and digit Mrgprd$^+$ nociceptors, while nociceptors from other regions of the hindlimb (including dorsal hindpaw and proximal hindlimb) grow long central arbors located more laterally.

Since <1% of Mrgprd$^+$ neurons are traced in these mice, it remains possible that this round-vs.-long arbor regional distinction may be an artifact of sparse labeling. For example, if both types were found throughout the DH but were differentially enriched between regions, sparse labeling might only trace the most prevalent type in each location. Using increasing dosages of tamoxifen, we found that these arbor types occupy mutually exclusive zones of the DH (*Figure 5*). The maintained segregation of long and round arbors in the DH despite the increased number of AP$^+$ DRG neurons indicates that these arbor morphologies represent a true somatotopic distinction among Mrgprd$^+$ non-peptidergic nociceptors.

## Neighboring non-peptidergic nociceptors highly overlap in the skin and spinal cord

The axonal arbors of some somatosensory neurons have a non-overlapping arrangement between neighbors ('tiling') in the body wall of the fly and zebrafish (*Grueber et al., 2002*; *Sagasti et al., 2005*). To determine if mammalian non-peptidergic nociceptive arbors tile in the skin, we generated double knock-in *Mrgprd$^{CreERT2/EGFPf}$*; *Rosa$^{tdTomato}$* mice. After low-dose tamoxifen treatment, sparsely labeled Mrgprd$^+$ neurons in these mice express tdT while the entire population expresses EGFP. The arbor fields of individual double tdT$^+$/EGFP$^+$ neurons are always co-innervated by EGFP-only$^+$ fibers in both hairy and glabrous skin (*Figure 6A–D*), indicating that peripheral arbors of neighboring non-peptidergic nociceptors do not tile but instead overlap extensively. In hairy skin, sparse labeled tdT$^+$ neurites co-innervated hair follicles with EGFP-only$^+$ fibers, indicating that multiple Mrgprd$^+$ neurons can innervate a single hair follicle. Similarly, both round and long DH arbors of non-peptidergic nociceptors highly overlap with their neighbors in the DH (*Figure 6E and F*).

## Heightened signal transmission in the paw DH circuitry of Mrgprd$^+$ neurons

Next, given the striking somatotopic differences in the central arbor organization of Mrgprd$^+$ nociceptors, we asked whether we could find regional (plantar paw vs. trunk) differences in the transmission of sensory information at the dorsal horn. We generated *Mrgprd$^{CreERT2}$*; *Rosa$^{ChR2-EYFP}$* mice, which were treated with postnatal tamoxifen (>P10, *Figure 1*), to compare the synaptic transmission of these neurons between somatotopic regions in spinal cord slice recordings. Comparable expression of ChR2-EYFP, as measured by the native fluorescence intensity, was found for Mrgprd$^+$ neurons in thoracic (T9-T12) and hind limb-level (L3-L5) DRGs (*Figure 7—figure supplement 1A–C*). We first used *Mrgprd$^{CreERT2/+}$*; *Rosa$^{ChR2-EYFP/ChR2-EYFP}$* mice (mice homozygous for the *Rosa$^{ChR2-EYFP}$* allele) to perform in vitro whole-cell patch-clamp recordings of layer II interneurons located in the

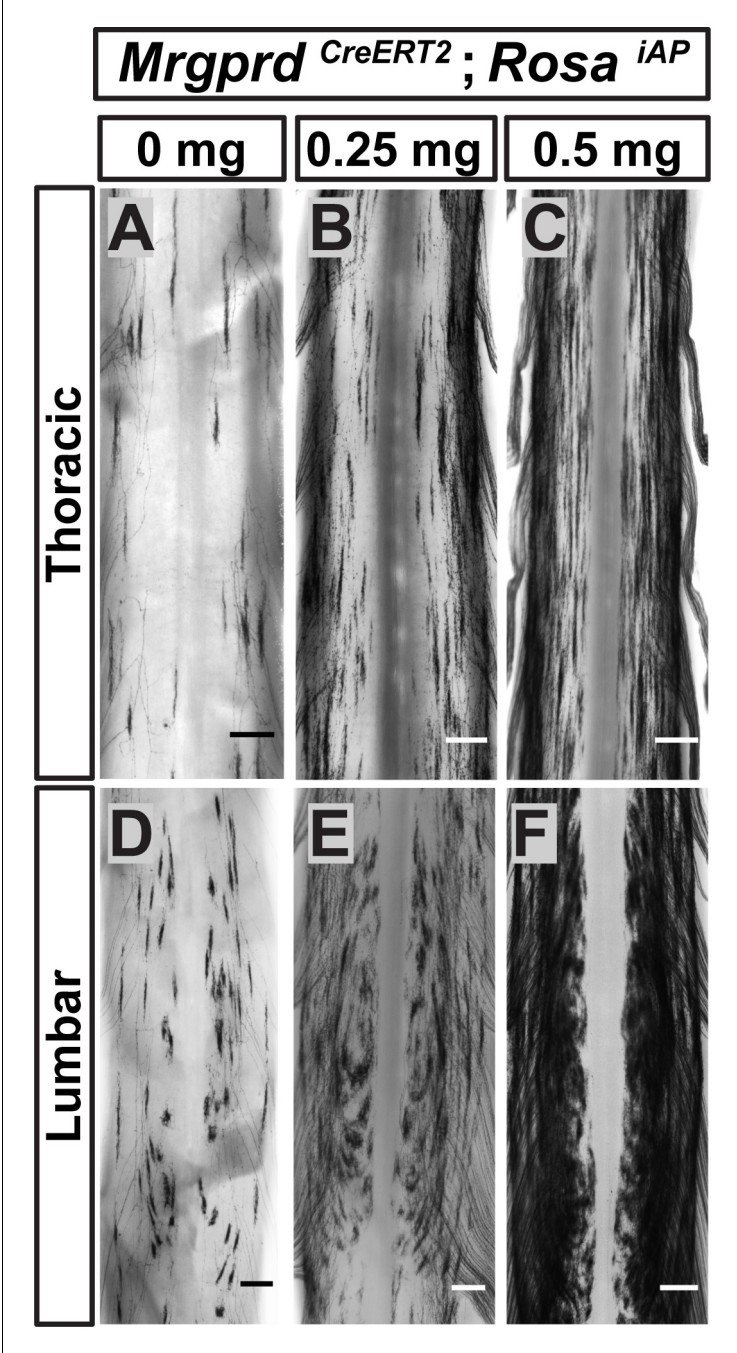

**Figure 5.** Non-peptidergic nociceptor labeling with increasing densities reveals somatotopic organization of Mgprd[+] central arbors. (**A–F**) AP staining of *Mrgprd^{CreERT2}; Rosa^{iAP}* thoracic (**A–C**) and lumbar (**D–F**) spinal cords that received prenatal 0 mg (**A** and **D**), 0.25 mg (**B** and **E**) or 0.5 mg (**C** and **F**) prenatal tamoxifen. Even with increased labeling densities, round and long arbors occupy exclusive zones of the DH. *n* = 3 animals per treatment. Scale bars = 250 μm.

DOI: https://doi.org/10.7554/eLife.29507.015

territory innervated by ChR2-EYFP[+] fibers in transverse spinal cord sections (*Figure 7A*). Light-evoked excitatory postsynaptic currents (EPSC_Ls) in these neurons could be differentiated into monosynaptic or polysynaptic responses based on latency, jitter, and response failure rate during 0.2 Hz blue light stimulation (*Figure 7B and C*) (*Cui et al., 2016*). Almost all recorded neurons in

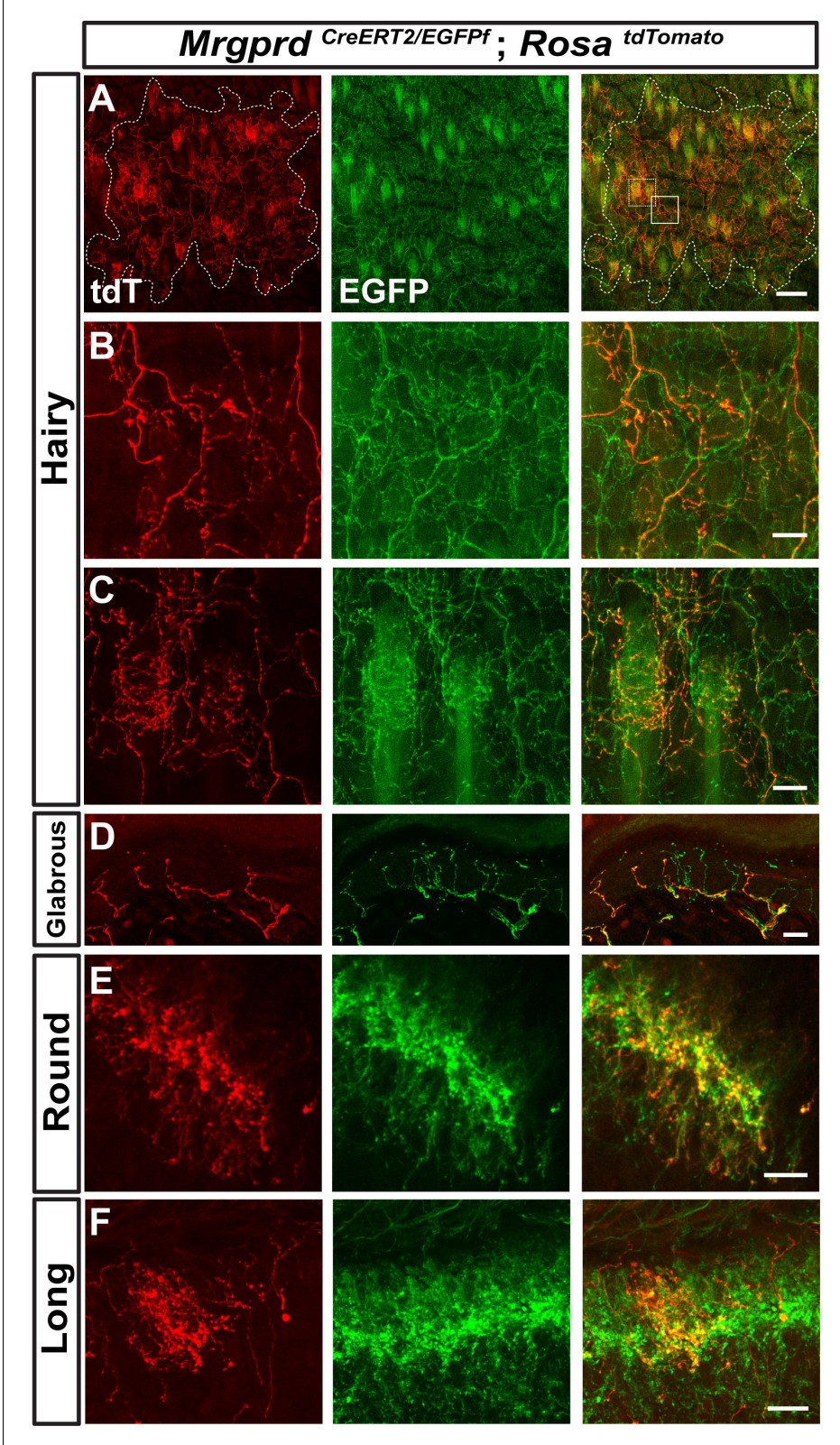

**Figure 6.** Neighboring non-peptidergic nociceptors overlap extensively in the skin and spinal cord. (**A–C**) Whole mount immunostaining of *Mrgprd*$^{CreERT2/EGFPf}$; *Rosa*$^{tdTomato}$ (0.5 mg tamoxifen at E16.5) hairy skin with anti-GFP and anti-RFP antibodies. The terminal field of one non-peptidergic nociceptor is labeled with tdT, as outlined in A. B and C show higher magnification views of the regions boxed in A (solid line = B, dotted line = C). Innervation of
*Figure 6 continued on next page*

*Figure 6 continued*

hair follicles is shown in C. (**D**) Immunostaining of a section of glabrous skin. (**E** and **F**) Immunostaining of medial cervical (**D**) and thoracic (**E**) spinal cord sections, showing sparse labeled round terminal and long arbors. Scale bars = 100 µm (**A**), 20 µm (**B–F**).

DOI: https://doi.org/10.7554/eLife.29507.016

both medial and lateral lumbar DH showed EPSC$_L$s (*Figure 7D*, *Figure 7—source data 1*), with most cells (14/17 = 82.4% in medial lumbar, 14/16 = 86.5% in lateral lumbar) showing monosynaptic EPSC$_L$s in both locations. This indicates that the majority of DH neurons in this innervation territory receive direct Mrgprd$^+$ input, and that the incidence receiving direct input is equivalent for medial lumbar and lateral lumbar DH.

Given the very high level of ChR2 expression in these mice, any potential difference between the medial and lateral lumbar spinal cord may be masked by a 'ceiling' effect. We therefore halved the genetic dosage of *Rosa$^{ChR2-EYFP}$* by taking slices from *Mrgprd$^{CreERT2}$; Rosa$^{ChR2-EYFP/+}$* mice (heterozygous for the *Rosa$^{ChR2-EYFP}$* allele). *Rosa$^{ChR2-EYFP}$* hetereozygous DRGs have a ~ 40% reduction in ChR2-EYFP protein compared to *Rosa$^{ChR2-EYFP}$* homozygous DRGs based on Western blotting (*Figure 7—figure supplement 1D and E*). Interestingly, we saw a dramatic difference when comparing the medial and lateral lumbar DH of double heterozygous mice. Medial lumbar DH neurons showed

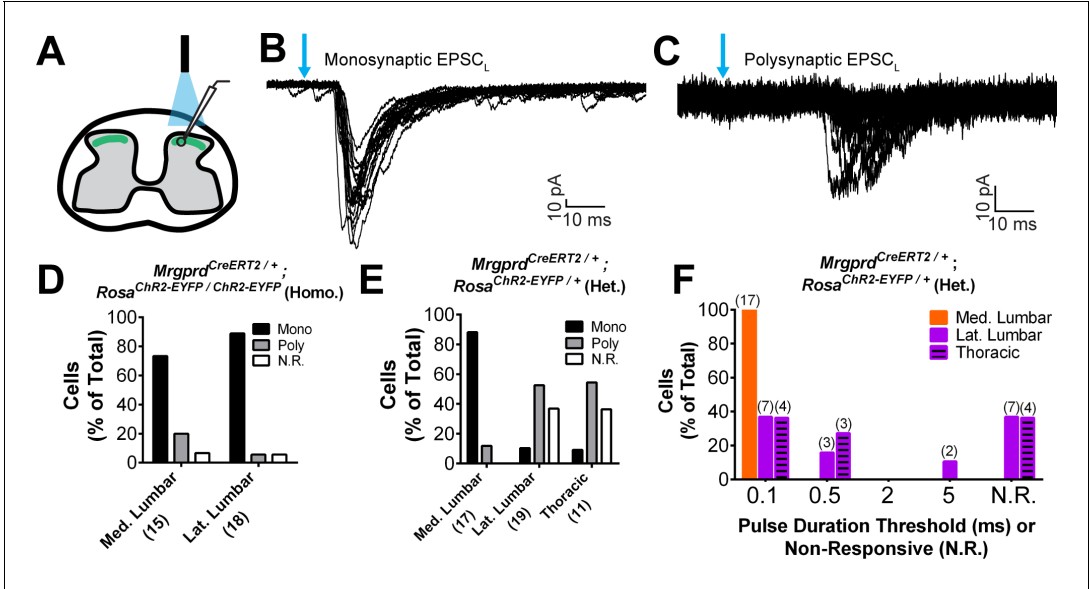

**Figure 7.** Plantar paw circuits show a heightened signal transmission in the dorsal horn. (**A**) Illustration of spinal cord slice recording from *Mrgprd$^{CreERT2}$; Rosa$^{ChR2-EYFP}$* mice (P14-P21 tamoxifen) using optical stimulation. Neuron cell bodies located in the territory innervated by EYFP$^+$ fibers were chosen for recording. (**B** and **C**) Monosynaptic (**B**) and polysynaptic (**C**) light-induced EPSC (EPSC$_L$) traces recorded from layer II neurons during 0.2 Hz light stimulation (overlay of 20 traces). Light pulses indicated by blue arrows, scale bars shown in lower right. (**D**) In *Mrgprd$^{CreERT2 / +}$; Rosa$^{ChR2-EYFP/ChR2-EYFP}$* homozygous slices, similar incidences of light-responsive neurons were found in medial and lateral lumbar regions. (**E**) In *Mrgprd$^{CreERT2 / +}$; Rosa$^{ChR2-EYFP/+}$* heterozygous slices, a much higher incidence of light-responsive neurons was seen in medial lumbar compared to lateral lumbar or medial thoracic circuits. (**F**) Frequency distribution of threshold light pulse durations required for eliciting EPSC$_L$s among cells in E. Among responsive cells, postsynaptic neurons in lateral lumbar and thoracic regions require longer pulse durations to be activated compared to those in medial lumbar region. Cell (n) numbers indicated in parentheses in x-axis labels in D and E and above bars in F.

DOI: https://doi.org/10.7554/eLife.29507.017

The following source data and figure supplement are available for figure 7:

**Source data 1.** Summary of incidences of light-induced excitatory postsynaptic current (EPSC$_L$) responses recorded from layer II neurons in *Mrgprd$^{CreERT2}$; Rosa$^{ChR2-EYFP}$* homozygous and heterozygous mice.

DOI: https://doi.org/10.7554/eLife.29507.019

**Figure supplement 1.** *Rosa$^{ChR2-EYFP}$* expression levels.

DOI: https://doi.org/10.7554/eLife.29507.018

a much higher incidence of light responses than lateral lumbar neurons, with a ~ 9 fold higher (15/17 = 88.2% vs. 2/19 = 10.5%) incidence of monosynaptic $EPSC_L$s over lateral lumbar neurons (*Figure 7E*, *Figure 7—source data 1*). Similar to the lateral lumbar DH, the incidence of monosynaptic $EPSC_L$s of layer II neurons in the medial and lateral thoracic region are also very low (1/11 = 9.1%) (*Figure 7E*, *Figure 7—source data 1*). Moreover, even among responsive neurons, the pulse duration threshold required to elicit $EPSC_L$s was much longer in lateral lumbar and thoracic DH neurons compared to medial lumbar neurons (*Figure 7F*). These results showed a lower threshold for light-triggered excitatory currents in medial lumbar compared to lateral lumbar and thoracic DH neurons, suggesting a heightened signal transmission in plantar paw circuits. We repeated these recordings in sagittal spinal cord slices and similarly found a higher $EPSC_L$ incidence in medial lumbar compared to lateral lumbar or thoracic DH (*Figure 7—source data 1*). This confirms that these differences are not caused by the different orientation of nociceptors in round vs. long arbor regions. Taken together, our results show that, while most DH neurons in the Mrgprd$^+$ innervation territory receive direct Mrgprd$^+$ input, the overall signal transmission is heightened in plantar paw compared to trunk nociceptive circuits. This heightened transmission was seen specifically at the level of nociceptor-to-DH neuron connections, and it correlates closely with the region-specific organization of Mrgprd$^+$ central arbors.

## Plantar paw Mrgprd$^+$ nociceptors have a lower stimulation threshold to induce avoidance behaviors

Finally, we asked whether the anatomical and physiological region-specific differences we identified are correlated with functional differences in a freely behaving animal. To activate ChR2 in skin-innervating nociceptors of behaving mice, we stimulated *Mrgprd$^{CreERT2}$*; *Rosa$^{ChR2-EYFP/ChR2-EYFP}$* (M) and *Rosa$^{ChR2-EYFP/ChR2-EYFP}$* (C) mice (P10-17 tamoxifen treatment) at both the paw and upper-thigh leg skin (*Figure 8B and E*) with either 473 nm blue light or 532 nm green light as a negative control. We only used ChR2 homozygous mice in these behavior assays because ChR2 heterozygous mice do not show any obvious response to peripheral optogenetic stimulation (data not shown). High levels of ChR2-EYFP were expressed in peripheral neurites in both plantar paw and upper leg skin (*Figure 8A and D*). Consistent with the AP labeling (*Figure 3*), upper leg skin shows a much higher density of EYFP$^+$ neurites than the paw glabrous skin (*Figure 8—figure supplement 1A–C*). In addition, ChR2-EYFP$^+$ neurites in paw glabrous skin terminate much farther from the surface than leg hairy skin neurites (*Figure 8—figure supplement 1A,B,D*) due to the thicker outer *stratum corneum* layer in glabrous skin.

We first stimulated the paw skin of both groups of mice with green light (5 mW, 10 Hz, sine wave) and observed no avoidance behavior such as paw withdrawal (*Figure 8C*, *Videos 1* and *2*). When we stimulated both groups of mice with 1 mW of blue light (10 Hz, sine wave), control mice did not respond, while 12.5% of *Mrgprd$^{CreERT2}$*; *Rosa$^{ChR2-EYFP}$* mice displayed light-induced paw withdrawal (*Figure 8B and C*). When stimulated with 5 mW blue light (10 Hz, sine wave), 100% of *Mrgprd$^{CreERT2}$*; *Rosa$^{ChR2-EYFP}$* mice displayed clear light-induced paw withdrawal. Control mice still did not respond (*Figure 8B and C*, *Figure 8—figure supplement 1F*, *Videos 3* and *4*).

Strikingly, when we activated Mrgprd$^+$ neurites in the shaved upper-thigh skin, neither control nor *Mrgprd$^{CreERT2}$*; *Rosa$^{ChR2-EYFP}$* mice responded to 5 mW blue light (*Figure 8E and F*, *Figure 8—figure supplement 1F*, *Videos 5* and *6*). Rather, to observe a fully penetrant avoidance behavior response in the upper-thigh of *Mrgprd$^{CreERT2}$*; *Rosa$^{ChR2-EYFP}$* mice, the blue light power intensity had to be increased to 10 or 20 mW (2–4 times higher than the requirement in the paw) (*Figure 8E–G*, *Videos 7 and 8*). When taking the temporal delay of avoidance responses (*Figure 8H*) into consideration, 20 mW intensity blue light is required at the leg to trigger responses comparable to 5 mW intensity stimulation of the paw. In short, the light intensity threshold required to trigger an avoidance response is significantly lower in the paw compared to the upper-limb of *Mrgprd$^{CreERT2}$*; *Rosa$^{ChR2-EYFP}$* mice.

## Discussion

In this study, we identified a novel somatotopic organization in the central arbors of mammalian Mrgprd$^+$ nociceptors, which is very well correlated with a regional increase in the sensitivity of paw nociceptive circuits to external stimuli. Our results suggest a model (*Figure 8I*), in which the wider

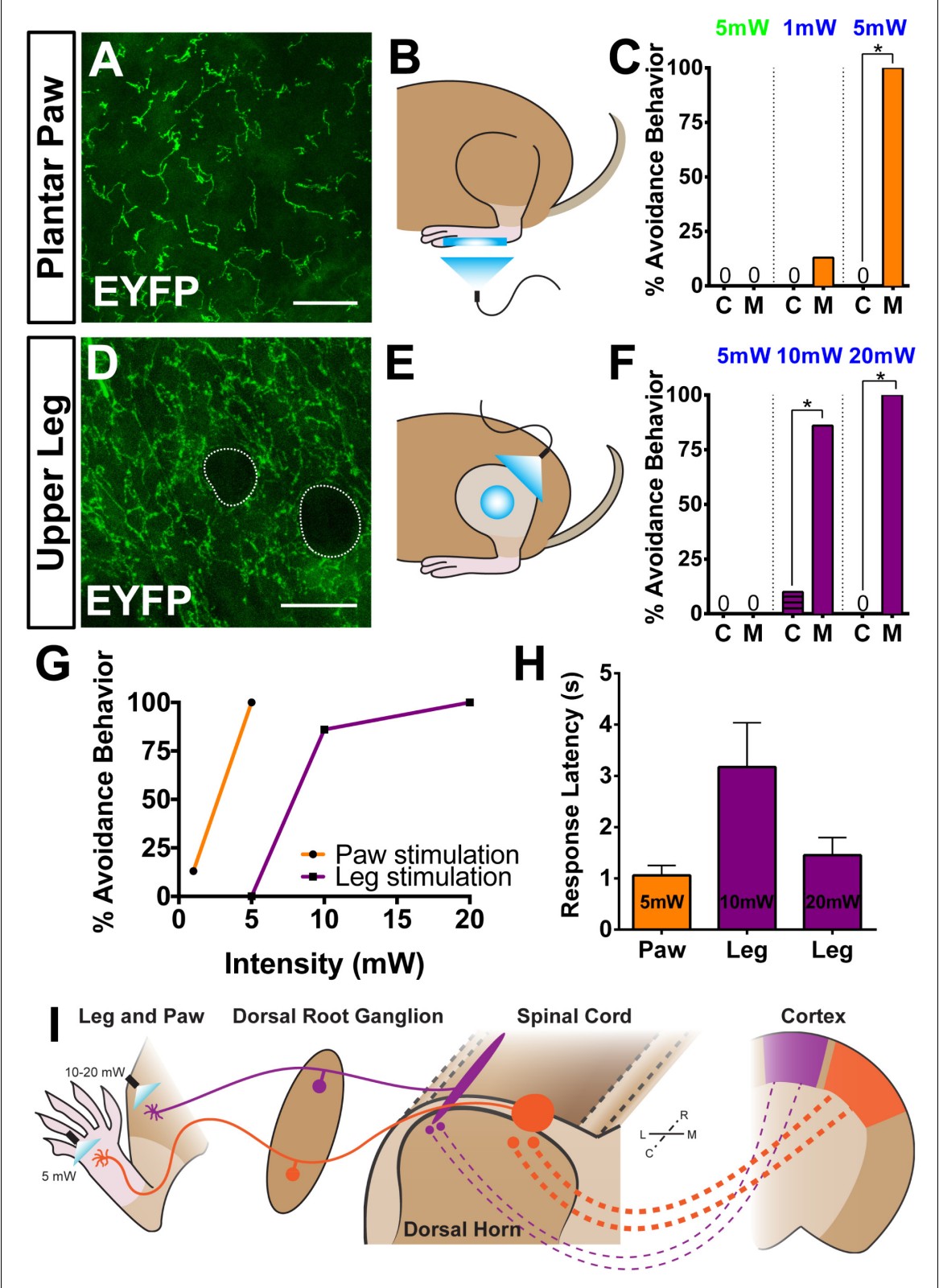

**Figure 8.** Peripheral optogenetic activation of Mrgprd$^+$ nociceptors reveals regional differences in optical threshold required to elicit withdrawal responses. (A–C) Optical stimulation in the paw. (A) Representative whole-mount immunostaining of plantar paw skin in *Mrgprd*$^{CreERT2 / +}$; *Rosa*$^{ChR2-EYFP/ChR2-EYFP}$ mice, *n* = 3 mice. (B) Schematic of light placement on paw skin, see videos. (C) Histogram showing percentage of mice displaying aversive responses to 5 mW green light and 1 or 5 mW blue light to littermate control (C) and *Mrgprd*$^{CreERT2 / +}$; *Rosa*$^{ChR2-EYFP/ChR2-EYFP}$(M). *n* = 6–10 for each

Figure 8 continued

genotype with 1–2 trials per mouse. *p<0.001 Chi-square test. (D–F) Optical stimulation in the leg. (D) Representative whole-mount immunostaining of upper leg skin in *Mrgprd*[CreERT2 / +]; *Rosa*[ChR2-EYFP/ChR2-EYFP] mice, n = 3 mice. Dotted lines outline hair follicles. (E) Schematic of light placement on hair-shaven leg skin, see videos. (F) Histogram showing percentage of mice displaying aversive responses to 5, 10, or 20 mW blue light at the leg (see above panel C for genotype description and statistical analyses). (G) A lower activation threshold is required for paw versus leg skin nociceptors. (H) Temporal delay time (seconds) from light onset to the first aversive behavior with 5, 10, or 20 mW blue light in paw or leg of *Mrgprd*[CreERT2 / +]; *Rosa*[ChR2-EYFP/ChR2-EYFP] mice. Error bars represent SEM. (I) Model showing somatotopic organization of mammalian nociceptive circuitry. Distinct central arbor morphologies ('round versus long') of Mrgprd[+] non-peptidergic nociceptors are observed in the medial versus lateral dorsal spinal cord, which correlates well with regional peripheral sensitivity and cortical representation. Scale bars = 50 μm.

DOI: https://doi.org/10.7554/eLife.29507.020

The following figure supplement is available for figure 8:

**Figure supplement 1.** In vivo optogenetic peripheral stimulation.

DOI: https://doi.org/10.7554/eLife.29507.021

mediolateral spread of plantar paw nociceptor central arbors could facilitate 'afferent magnification" (*Catania and Kaas, 1997*) in downstream CNS circuits and facilitate heightened pain sensitivity of the plantar paw. Remarkably, two features of mouse non-peptidergic nociceptors revealed by our study, the peripheral neurite density distribution and the heightened sensitivity of pain processing in the distal limb, are consistent with findings in humans (*Mancini et al., 2012*; *Mancini et al., 2013*; *Mancini et al., 2014*). Therefore, the organizational mechanisms we discovered in mice are likely to be conserved in humans, which provides a possible explanation for the human 'pain fovea'.

## *Mrgprd*[CreERT2] allows for specific targeting of adult Mrgprd[+] nociceptors

Adult mice have two functionally distinct DRG neuronal populations expressing Mas-related gene product receptor (*Mrgpr*) genes. One expresses *Mrgpra*, *Mrgprb*, and *Mrgprc* genes and the other only expresses *Mrgprd* (*Liu et al., 2012*; *Liu et al., 2008*; *Dong et al., 2001*; *Han et al., 2013*; *Liu et al., 2009*; *Liu et al., 2007*; *Zylka et al., 2003*). Mrgpra/b/c[+] neurons also transiently express *Mrgprd* during early development (*Liu et al., 2008*). To specifically target Mrgprd[+] neurons, we generated a new inducible *Mrgprd*[CreERT2] mouse line that allows for temporally controlled recombination. We demonstrated that postnatal (P10 or later) tamoxifen treatment of *Mrgprd*[CreERT2] mice specifically targets Mrgprd[+] but not Mrgpra/b/c[+] neurons (*Figure 1*).

To our advantage, we also found that sparse recombination in untreated *Mrgprd*[CreERT2]; *Rosa*[iAP] mice is very specific for Mrgprd[+] neurons (~94% AP[+] neurons are Mrgprd[+], *Figure 2I*). We noticed that most of this random recombination likely occurs postnatally, as untreated *Mrgprd*[CreERT2]; *Rosa*[iAP] tissue at P7 or younger shows no AP[+] neurons (data not shown). This temporal delay of recombination, along with the fact that there are many more Mrgprd[+] than Mrgpra/b/c[+] neurons (*Liu et al., 2008*; *Dong et al., 2001*; *Liu et al., 2009*; *Liu et al., 2007*), likely contributes to the specificity of sparse AP labeling in these mice.

In this paper, we used prenatal tamoxifen treatment of *Mrgprd*[CreERT2] mice for two experiments: increasing density labeling (*Figure 5*) and *Mrgprd*[CreERT2/EGFPf]; *Rosa*[tdTomato] labeling to show nociceptor overlap (*Figure 6*). In these experiments, both Mrgprd[+] and Mrgpra/b/c[+] neurons could be targeted. For the overlap experiment, neurites were chosen that had both red (*Mrgprd*[CreERT2]; *Rosa*[tdTomato]) and green (*Mrgprd*[EGFPf]) fluorescence at 3pw, indicating these were Mrgprd[+] neurites. For the increased density labeling experiment, even with high dosage (population-level) prenatal tamoxifen

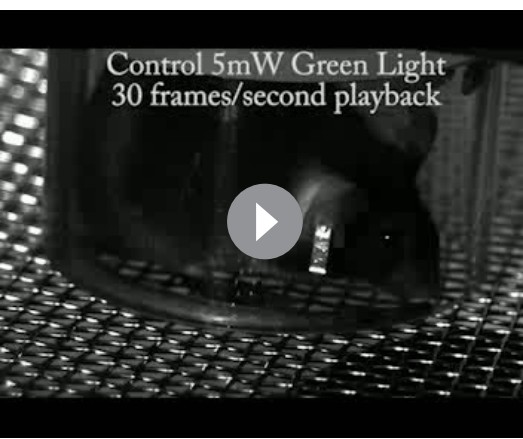

**Video 1.** High speed video recording of green laser light (5 mW) plantar paw stimulation of Control (*Rosa*[ChR2-EYFP]) mice.
DOI: https://doi.org/10.7554/eLife.29507.022

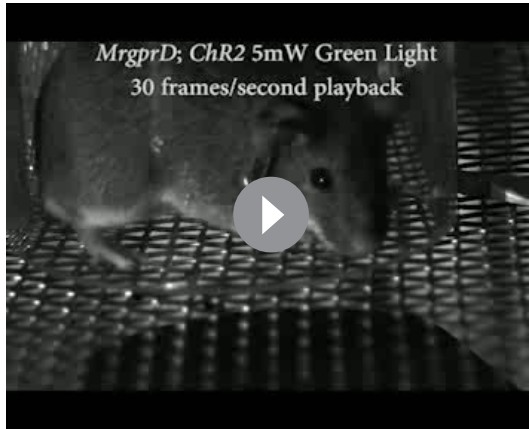

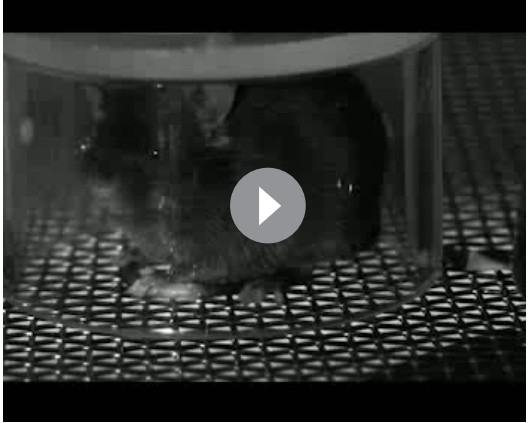

**Video 2.** High speed video recording of green laser light (5 mW) plantar paw stimulation of *Mrgprd*[CreERT2]; *Rosa*[ChR2-EYFP] mice.
DOI: https://doi.org/10.7554/eLife.29507.023

**Video 3.** High speed video recording of blue laser light (5 mW) plantar paw stimulation of Control mice.
DOI: https://doi.org/10.7554/eLife.29507.024

treatment of this line, Mrgpra/b/c[+] neurons make up <20% of the total cells labeled (*Figure 1—figure supplement 2*). Given the nature of this experiment, we believe that our interpretation is not confounded by this issue.

## Mrgprd[+] nociceptors show no obvious peripheral features for heightened pain sensitivity in the paw

Though previous studies have traced various cutaneous somatosensory neurons, no systematic characterization of nociceptor morphology across the somatotopic map has been performed. In the periphery, sparse tracing of *Pou4f1* expressing somatosensory neurons (which includes almost all somatosensory DRG neuron classes [*Badea et al., 2012*]) in the back hairy revealed a 'bushy ending' morphological type (*Wu et al., 2012*). The authors suggested these terminals might correspond to C-fiber nociceptors or thermoceptors. In addition, Mrgprb4[+] and TH[+] C fibers, which mainly mediate light touch but not pain, and innervate hairy skin only, have been genetically traced and analyzed (*Li et al., 2011*; *Liu et al., 2007*). Mrgprb4[+] neurons innervate the skin in large patches of free terminals (*Liu et al., 2007*), while TH[+] neurons form lanceolate endings around hair follicles (*Li et al., 2011*). To our knowledge, single-cell tracing has not previously been performed for any C-fiber nociceptors innervating the glabrous skin.

Mrgprd[+] nociceptors innervate both hairy and glabrous skin but not deep tissues (*Zylka et al., 2005*), making this population ideal for analysis of somatotopic differences. Our *Mrgprd*[CreERT2] tracing reveals that Mrgprd[+] nociceptors display a bushy-ending morphology in hairy skin and thickened endings in the epidermis in the glabrous skin (the plantar paw and finger tips) (*Figure 3*, *Figure 3—figure supplement 1*). Interestingly, individual Mrgprd[+] afferents in hairy skin innervate both the hair follicle and the interfollicular skin (*Figure 3*, *Figure 3—figure supplement 1*, *Figure 6*). Recently, a separate population of afferents that express CGRP and form circumferential endings around the deep hair follicle (deeper than Mrgprd[+] endings) were identified as mediators of pain upon hair pulling (*Ghitani et al., 2017*).

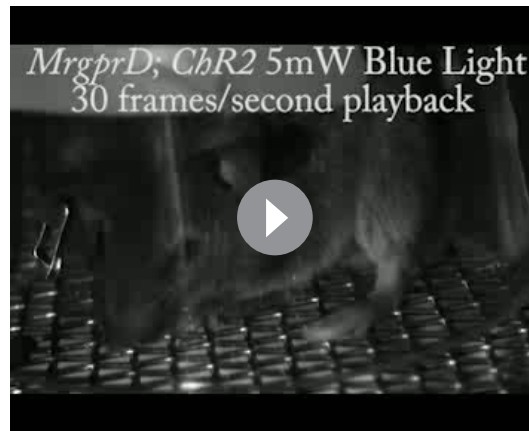

**Video 4.** High speed video recording of blue laser light (5 mW) plantar paw stimulation of *Mrgprd*[CreERT2]; *Rosa*[ChR2-EYFP] mice.
DOI: https://doi.org/10.7554/eLife.29507.025

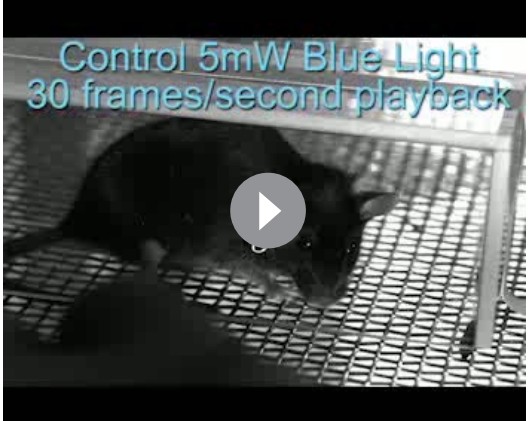

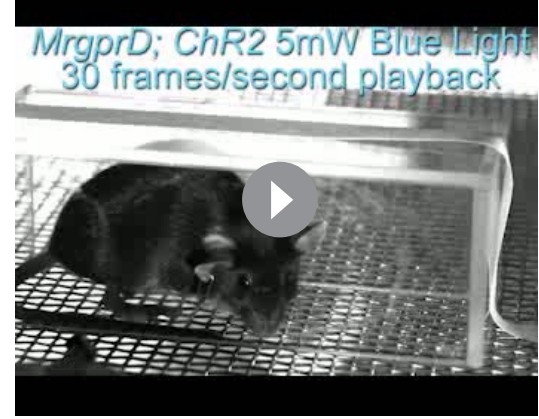

**Video 5.** High speed video recording of blue laser light (5 mW) shaved upper leg stimulation of Control mice.
DOI: https://doi.org/10.7554/eLife.29507.026

**Video 6.** High speed video recording of blue laser light (5 mW) upper leg stimulation of *Mrgprd*$^{CreERT2}$; *Rosa*$^{ChR2-EYFP}$ mice.
DOI: https://doi.org/10.7554/eLife.29507.027

Our single-cell tracing suggests that a single Mrgprd$^+$ afferent may signal both hair pulling as well as mechanical noxious stimuli applied to the skin surface. Further analysis would be required to test this possibility. Very rarely, we found 'free ending' terminals that lack thickened epidermal ending (*Figure 3—figure supplement 1*). These match the morphology and size of Mrgprb4$^+$ light touch neurons (*Liu et al., 2007*), possibly indicating very rare labeling of this subset. The fact that <2% of hairy skin terminals displayed this morphology further supports the specificity of *Mrgprd*$^{CreERT2}$ sparse recombination.

We found that (1) the density of Mrgprd$^+$ C fiber nociceptive neurites is similar or slightly lower in paw (including digit tips) compared to trunk skin, (2) Mrgprd$^+$ neuron density is lower in the plantar paw compared to the upper hind limb skin, and (3) paw and trunk individual terminals are comparable in innervation area (*Figure 3*). This last result contrasts with an earlier study that performed single-fiber recordings of human mechanically responsive C-fiber units, which saw smaller receptive fields in the distal leg (*Schmidt et al., 1997*). This discrepancy could be caused by differences between species, techniques (physiological vs. direct anatomic tracing), or the composition of neurons that were analyzed (the previous study presumably recorded from multiple molecular classes). In short, in our analysis of mouse Mrgprd$^+$ nociceptors, we did not find any obvious peripheral mechanism that might readily explain the heightened pain acuity of the distal limbs seen in human subjects or the increased sensitivity of the mouse plantar paw to optogenetic skin stimulation (*Figure 8*).

### Mrgprd$^+$ nociceptors display region-specific organization of central arbors

Classic studies have characterized the single-cell central arbors of C-fiber afferents using Golgi staining or backfill techniques and described them as longitudinally-oriented 'thin sheets' that are short in the mediolateral axis and extended in the rostrocadual axis (*Sugiura et al., 1986*; *Sugiura et al., 1989*; *Scheibel and Scheibel, 1968*). This description corresponds well to the 'long arbors' we found in DH zones representing proximal limb and trunk regions (*Figure 4*). The central terminals of Mrgprb4$^+$ and TH$^+$ C-fiber

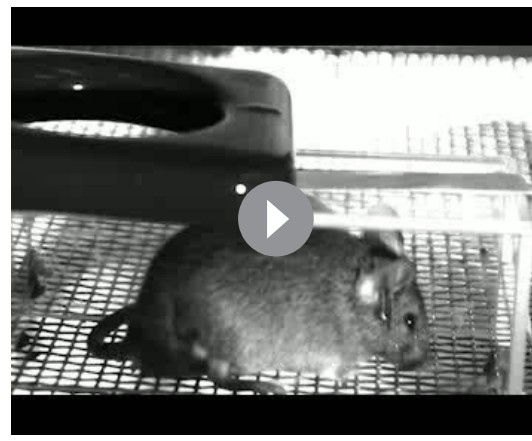

**Video 7.** High speed video recording of blue laser light (20 mW) upper leg stimulation of Control mice.
DOI: https://doi.org/10.7554/eLife.29507.028

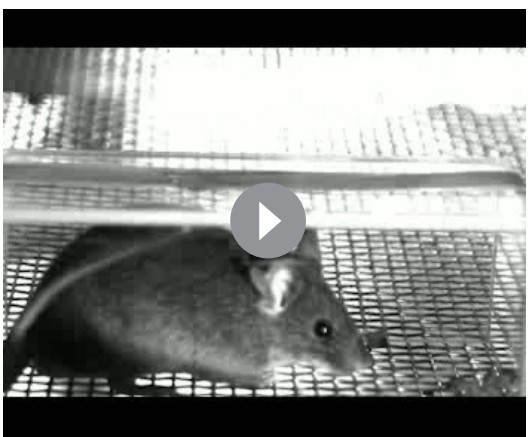

**Video 8.** High speed video recording of blue laser light (20 mW) upper leg stimulation of *Mrgprd^CreERT2^; Rosa^ChR2-EYFP^* mice.

DOI: https://doi.org/10.7554/eLife.29507.029

light touch neurons also match this long terminal morphology, consistent with the fact that these classes only innervate hairy skin (*Li et al., 2011*; *Liu et al., 2007*). Nevertheless, a systematic comparison of nociceptive central terminals across the entire somatotopic map has not been conducted.

Here, we found regionally distinctive central arbor organization among Mrgprd⁺ nociceptors; those innervating the distal limbs, tail, anogenital skin, and the head/face display round central terminals, while those innervating the trunk hairy skin display long and thin central arbors (*Figure 4*). Given that neurons in the medulla and sacral spinal cord display round arbors (*Figure 4* and *Figure 4—figure supplement 1C*), this morphological difference does not correlate with hairy vs. glabrous skin regions but instead seems to correlate with regions located at the extremities. In addition, upon optogenetic activation of Mrgprd⁺ central terminals, we saw a close correlation between region-specific central arbor type (round vs. long) and signal transmission strength. In *Mrgprd^CreERT2^;Rosa^ChR2-EYFP^* heterozygous mice, lateral lumbar and thoracic circuits showed similar post-synaptic response incidences/stimulus duration thresholds, whereas medial lumbar circuits had a much higher response incidence and a lower stimulus duration threshold (*Figure 7*). While we cannot rule out a role for regional differences in the single-cell physiology of Mrgprd⁺ neurons (such activation threshold, transmitter release, etc.), this correlation suggests that region-specific central arbor structure could be a key contributor to the increased pain signal transmission of paw circuits. Taken together, we have uncovered a novel form of region-specific functional organization for mammalian nociceptors.

Previous psychophysical studies that defined the 'pain fovea' in humans used heat nociceptive stimuli. Thermal pain in mice is primarily mediated through a separate population of afferents (CGRP⁺ peptidergic nociceptors) from the Mrgprd⁺ fibers that we studied (*Cavanaugh et al., 2009*; *McCoy et al., 2012*). This suggests that region-specific organization could exist in peptidergic fibers as well. However, whether they also display somatotopic central arbor differences like Mrgprd⁺ nonpeptidergic nociceptors remains to be determined. In addition, some previous studies suggest that the third-order DH collaterals of Aβ mechanoreceptors are also wider (in the mediolateral axis) in the medial lumbar enlargement compared to other DH regions (*Brown et al., 1991*; *Shortland et al., 1989*; *Millecchia et al., 1991*). However, the primary signal transmission for discriminative information occurs in the dorsal column nuclei but not the dorsal spinal cord. Thus, it is currently less clear whether this central arbor morphological difference contributes to differential touch sensitivity.

Interneurons in DH layer II are morphologically and physiologically heterogeneous, and several interneuron subtypes have been defined (*Grudt and Perl, 2002*; *Lu and Perl, 2005*; *Lu and Perl, 2003*). Mrgprd⁺ fibers synapse on most identified classes of layer II interneurons (*Wang and Zylka, 2009*) and signal to pain pathway projection neurons located outside of layer II through polysynaptic pathways (*Lu and Perl, 2005*). However, past work on these circuits oftentimes focused on lumbar enlargement spinal cord. It is therefore unclear if there are systematic somatotopic differences in the subtypes or organization of DH interneuron circuits, and whether the different morphological central arbor types identified by our study may contact the same or different downstream pathways.

## Increased sensitivity of mouse paw pain circuits to external input

Finally, we determined the stimulus threshold (laser power) required to trigger avoidance behaviors of freely behaving *Mrgprd^CreERT2^; Rosa^ChR2-EYFP^* mice upon paw or upper leg skin light stimulation. Our experiments show a clear heightened sensitivity of plantar-paw-innervating Mrgprd⁺ nociceptors to stimulation (*Figure 8*, *Figure 8—figure supplement 1*). Interestingly, our use of peripheral

optogenetic stimulation likely bypasses the effects of some peripheral parameters, such as the mechanical or thermal conduction properties of the skin or the expression of molecular receptors, on the sensation of natural stimuli. Further, given that Mrgprd$^+$ neurites terminate farther from the skin surface (*Figure 8—figure supplement 1*), it is unlikely that the laser stimulus has more ready access to Mrgprd$^+$ terminals in plantar paw skin. Nevertheless, we cannot rule out the contribution of other regional differences among the peripheral terminals of these neurons that we did not measure.

Collectively, our results suggest that region-specific Mrgprd$^+$ central arbor organization could magnify the representation of the plantar paw within pain circuits to contribute to heightened pain sensitivity (*Millecchia et al., 1991*) (*Figure 8I*). This central organization mechanism could allow for the sensitive detection of potential harmful stimuli by these skin areas, despite a lower density in the periphery. This finding is relevant for pain research using rodent models, which has historically relied heavily on pain assays in plantar paw/medial lumbar circuits (*Le Bars et al., 2001*). Given our findings, it would be interesting to examine whether region-specific differences exist in the molecular and physiological pathways of acute and/or chronic pain models. Such work could be informative for the translation of preclinical models to clinical treatment.

# Materials and methods

## Mouse strains

Mice were raised in a barrier facility in Hill Pavilion, University of Pennsylvania. All procedures were conducted according to animal protocols approved by Institutional Animal Care and Use Committee (IACUC) of the University of Pennsylvania and National Institutes of Health guidelines. *Rosa$^{tdTomato}$* (RRID:IMSR_JAX:007909), *Rosa$^{iAP}$* (RRID:IMSR_JAX:009253), *Rosa$^{ChR2-EYFP}$* (RRID:IMSR_JAX:012569), and *Mrgprd$^{EGFPf}$* (RRID:IMSR_TAC:tf0437) lines have been described previously (*Zylka et al., 2005*; *Badea et al., 2009*; *Madisen et al., 2010*; *Madisen et al., 2012*).

## Generation of *Mrgprd$^{CreERT2}$* mice

Targeting construct arms were subcloned from a C57BL/6J BAC clone (RP24-316N16) using BAC recombineering, and the CreERT2 coding sequence followed by a FRT-flanked neomycin-resistance selection cassette was engineered in-frame following the *Mrgprd* starting codon by the same approach (*Figure 1—figure supplement 1A*). The targeting construct was electroporated into a C57/129 hybrid (V6.5, RRID: CVCL_C865) mouse embryonic stem cell line by the Penn Gene Targeting Core. V6.5 cells were provided directly from the group that developed the line (*Eggan et al., 2001*) at passage 12 and then expanded in the Penn targeting vector core to passage 15, which was used for *Mrgprd$^{CreERT2}$* targeting. Cells were analyzed for correct morphology and chromosome number and were confirmed to be mycoplasma negative. ES clones were screened by PCR using primers flanking the 3' insertion site (*Figure 1—figure supplement 1C*). Positive clones were further screened using Southern blot with both internal and external probes (*Figure 1—figure supplement 1B*). The Penn Transgenic Core assisted in *Mrgprd$^{CreERT2}$* ES clone blastocyst injection and in the generation of chimeric mice, which were mated to a *Rosa$^{Flippase}$* line (RRID:MMRRC_007844-UCD) to excise the Neo cassette. Neo cassette-negative progeny (verified via PCR of genomic DNA) were mated to C57 (RRID:IMSR_JAX:000664 or CD1 (RRID:IMSR_CRL:22) mice to establish the line.

## Genetic labeling of mrgprd$^+$ nociceptors

To label Mrgprd$^+$ nociceptors, *Mrgprd$^{CreERT2}$* mice carrying the relevant reporter allele were treated with tamoxifen ((Sigma-Aldrich, St. Louis, MO, T5648) pre- or postnatally. For prenatal treatment, pregnant females were given tamoxifen along with estradiol (Sigma, E8875, at a 1:1000 mass estradiol: mass tamoxifen ratio) and progesterone (Sigma, P3972, at a 1:2 mass progesterone: mass tamoxifen ratio) in sunflower seed oil via oral gavage at E16.5-E17.5, when *Mrgprd* is highly expressed in mouse non-peptidergic nociceptors (*Chen et al., 2006*). For postnatal treatment, 0.5 mg tamoxifen extracted in sunflower seed oil was given via i.p. injection once per day from P10-P17 (or P14-P21 for spinal cord slice recording experiments, *Figure 7*). At least one week was given to drive recombination and reporter gene expression.

## Tissue preparation and histology

Procedures were conducted as previously described (*Fleming et al., 2012*). Briefly, mice used for immunostaining or AP staining were euthanized with $CO_2$ and transcardially perfused with 4% PFA/PBS, and dissected tissue (either skin or spinal cord and DRGs) was post-fixed for 2 hr in 4% PFA/PBS at 4° C. Tissue used for immunostaining was cryo-protected in 30% sucrose/PBS (4% overnight) before freezing. Mice used for in situ hybridization were euthanized and unfixed dissected tissue was frozen. Frozen glabrous skin and DRG/spinal cord sections (20–30 μm) were cut on a Leica CM1950 cryostat. Immunostaining of sectioned DRG, spinal cord, and glabrous skin tissue, whole mount skin immunostaining, and double fluorescence in situ hybridization was performed as described previously (*Fleming et al., 2012*; *Niu et al., 2013*). The following antibodies and dyes were used: rabbit anti-CGRP (ImmunoStar, Hudson, WI, Cat# 24112, RRID:AB_572217), rat anti-CK8 (DSHB, Iowa City, IA, Cat# TROMA-I, RRID:AB_531826), chicken anti-GFP (Aves Labs, Tigard, OR, Cat# GFP-1020, RRID:AB_10000240), rabbit anti-GFP (Thermo Fisher Scientific, Waltham, MA, Cat# A-11122 also A11122, RRID:AB_221569), conjugated IB4-Alex488 (Molecular Probes, Eugene, OR, Cat# I21411 also I21411, RRID:AB_2314662), rabbit anti-NF200 (Sigma, Cat# N4142, RRID:AB_477272), chicken anti-PAP (Aves Labs PAP, RRID:AB_2313557), chicken anti-peripherin (Aves Labs Cat# ABIN361364, RRID:AB_10785694), mouse anti-PKCγ (Innovative Research, Sharon Hill, PA, Cat# 13–3800, RRID:AB_86589), rabbit anti-RET (Immuno-Biological Laboratories, Minneapolis, MN, Cat# 18121, RRID:AB_2301042), rabbit anti-RFP (Clontech Laboratories, Inc., Mountain View, CA, Cat# 632496, RRID:AB_10013483). *Mrgprd*, *Mrgpra3*, and *Mrgprb4 in situ* probes were previously described (*Luo et al., 2007*).

Tissue (skin or spinal cord with attached DRGs) for whole mount AP colorimetric staining with BCIP/NBT substrate (Roche, Basel, Switzerland, 1138221001 and 11383213001) and for fluorescent staining with HNPP/FastRed substrate (Roche, 11758888001) was treated as previously described (*Niu et al., 2013*). Following AP colorimetric labeling, tissue was either cleared in BABB for imaging or sectioned using a VT1200S vibratome (Leica Microsystems, Nussloch, Germany) (200 μm), followed by BABB clearing for imaging. Fluorescent labeled DRGs co-stained using antibodies were cleared in glycerol and for imaging.

## Retrograde DiI labeling

DiI (Molecular Probes, Cat# D282, 1 μL, 0.2 mg/mL dissolved in DMSO then diluted 1:5 with PBS) was subcutaneously injected in the plantar hind paw or shaved ventral proximal hind limb of 4pw *Mrgprd^{EGFPf}* mice (each mouse received DiI at both locations on opposite sides, and side-location combinations were alternated between mice). 7 days after injection, mice were perfused and skin and L3-L5 DRGs were dissected. Skin was post-fixed and mounted in PBS for imaging. DRGs were post-fixed, cryoprotected and then serially cryosectioned through the whole ganglion, and sections were mounted and imaged for quantification.

## Electrophysiology

Spinal cord slices recordings were conducted as previously described (*Cui et al., 2011*). Basically, 4-6pw *Mrgprd^{CreERT2}*; *Rosa^{ChR2-EYFP/ChR2-EYFP}* or *Mrgprd^{CreERT2}*; *Rosa^{ChR2-EYFP/+}* mice were anesthetized with a ketamine/xylazine/acepromazine cocktail. Laminectomy was performed, and the spinal cord lumbar segments were removed and placed in ice-cold incubation solution consisting of (in mM) 95 NaCl, 1.8 KCl, 1.2 $KH_2PO_4$, 0.5 $CaCl_2$, 7 $MgSO_4$, 26 $NaHCO_3$, 15 glucose, and 50 sucrose, oxygenated with 95% $O_2$ and 5% $CO_2$, at a pH of 7.35–7.45 and an osmolality of 310–320 mosM. Sagittal or transverse spinal cord slices (300–500 μm thick) were prepared using a VT1200S vibratome ) and incubated in 34°C incubation solution for 30 min.

The slice was transferred to the recording chamber and continuously perfused with recording solution at a rate of 3–4 ml/min. The recording solution consisted of (in mM) 127 NaCl, 1.8 KCl, 1.2 $KH_2PO_4$, 2.4 $CaCl_2$, 1.3 $MgSO_4$, 26 $NaHCO_3$, and 15 glucose, oxygenated with 95% $O_2$ and 5% $CO_2$, at a pH of 7.35–7.45 and an osmolality of 300–310 mosM. Recordings were performed at RT. Spinal cord slices were visualized with an Olympus BX 61WI microscope (Olympus Optical, Tokyo, Japan), and the substantia gelatinosa (lamina II), which is a translucent band across the dorsal horn, was used as a landmark. Fluorescently labeled ChR2-EYFP terminals in the DH were identified by epifluorescence, and neurons in this innervation territory were recorded in the whole cell patch-

clamp configuration. Glass pipettes (3–5 MΩ) were filled with internal solution consisting of (in mM) 120 K-gluconate, 10 KCl, 2 MgATP, 0.5 NaGTP, 20 HEPES, 0.5 EGTA, and 10 phosphocreatine di (tris) salt at a pH of 7.29 and an osmolality of 300 mosM. All data were acquired using an EPC-9 patch-clamp amplifier and Pulse software (HEKA, Freiburg, Germany). Liquid junction potentials were not corrected. The series resistance was between 10 and 25 MΩ.

Light induced EPSCs ($EPSC_L$s) were elicited at a frequency of 2/min by 473 nm laser illumination (10 mW, 0.1–5 ms, Blue Sky Research, Milpitas, USA). Blue light was delivered through a 40X water-immersion microscope objective. Mono- or polysynaptic $EPSC_L$s were differentiated by 0.2 Hz light stimulation. We classified a connection as monosynaptic if the EPSC jitter (average standard deviation of the light-induced EPSCs latency from stimulation)<1.6 ms (*Wang and Zylka, 2009*), (*Doyle and Andresen, 2001*).

## Western blotting

Western blotting was performed as previously described (*Fleming et al., 2015*). Briefly, L3-L5 DRG protein lysates were prepared in 600 uL RIPA buffer (50 mM Tris, 150 mM NaCl, 1% NP-40, 0.5% sodium deoxycholate, 0.1% sodium dodecyl sulfate, pH = 8.0) with added protease inhibitors (Sigma, P8340), and mixed with equal parts 2X Sample buffer (0.125 M Tris, 20% glycerol, 4% SDS, 0.16% bromophenol blue, 10% 2-mercapatoethanol) before denaturing (100°C, 10 min). 10 uL of lysate samples were run on duplicate 4–15% mini-Protean TGX gels (Biorad, 456–1086), and both gels were transferred to nitrocellulose membranes. Membranes were blotted with either rabbit anti-GFP (1:2000) or rabbit anti-NF200 (1:2000), followed by AP-conjugated goat anti-rabbit secondary (1:5000, Thermo Fisher Scientific Cat# T2191, RRID:AB_11180336). AP was detected using CDP-Star (Thermo Scientific, T2218) and imaged with a Chemi-Blot System (BioRad).

## Optogenetic stimulation of Mrgprd[+] nociceptors in paw and leg skin

To induce light-evoked behavior in freely moving animals, we used P10-P17 tamoxifen treated $Mrgprd^{CreERT2}$; $Rosa^{ChR2-EYFP/ChR2-EYFP}$ mice and control littermates ($Rosa^{ChR2-EYFP/ChR2-EYFP}$) who were also tamoxifen treated, but lacked the Cre-driver. All tested animals were between 2–6 months old. An additional control was to shine 532 nm green laser light (Shanghai Laser and Optics Century, Shanghai, China, GL532T8-1000FC/ADR-800A) to the paw skin of both experimental groups. To induce light-evoked aversive behavior in Mrgprd[+] neurites in the paw skin, 1 or 5 mW of 473 nm blue light laser (Shanghai Laser and Optics Century, BL473T8-150FC/ADR-800A) was shined directly to the paw skin through a mesh bottom floor, with a cutoff time of 10 s. We tested different light waveforms and found that 10 Hz sine waveform pulsing gave the best behavior responses. Thus, we used this waveform for all our behavior tests.

To induce light-evoked aversive behavior in Mrgprd[+] neurites in the leg skin, first, all hair was removed from the leg with Nair hair remover under light 3% isoflourane anesthesia, and animals were given two days before being tested again. For leg stimulation, 1, 5, 10, or 20 mW of 473 nm blue light laser, with 10 Hz sine waveform pulsing, was shined directly to the leg skin. The cutoff time for this behavior assay is 10 s. For habituation to either paw or leg skin light stimulation, on the first two days, animals were habituated to the testing paradigm by being placed in the plexiglass testing chamber (11.5 × 11.5 × 16 cm) for 30 min each day. On the third testing day, animals were placed in the plexiglass testing chamber for 15 min prior to light stimulation. Green and blue light testing were performed on different days, but two weeks separated the paw and leg skin light stimulation. For all stimulation, the laser light was delivered via an FC/PC Optogenetic Patch Cable with a 200micrometer core opening (ThorLabs) and there was approximately 1 cm of space between the cable terminal and the targeted skin area. Light power intensity for each experiment was measured with a Digital Power Meter with a 9.5 mm aperture (ThorLabs). For leg skin stimulation, power intensity was only slightly impeded by the thin wall (0.02 cm) of the plexiglass holding chamber, as measured by the Digital Power Meter (*Figure 8—figure supplement 1E*).

To gain precise spatial and temporal resolution of behavior responses, we recorded behaving animals at 500 frames/second with a high-speed camera (FASTCAM UX100 800 K-M-4GB - Monochrome 800K with 4 GB memory) and attached lens (Nikon Zoom Wide Angle Telephoto 24–85 mm f2.8). With a tripod with geared head for Photron UX100, the camera was placed at a ~ 45 degree angle at ~1–2 feet away from the plexiglass holding chambers where mice performed behaviors. The

camera was maximally activated with far-red shifted 10 mW LED light that did not disturb animal behavior. All data were collected and annotated on a Dell laptop computer with Fastcam NI DAQ software that is designed to synchronize Photron slow motion cameras with the M series integrated BNC Data Acquisition (DAQ) units from National Instruments.

### Image acquisition and data analysis

Images were acquired either on a Leica DM5000B microscope (bright field with a Leica DFC 295 camera and fluorescent with a Leica 345 FX camera), on a Lecia SP5II confocal microscope (fluorescent), or on a Leica M205 C stereoscope with a Leica DFC 450 C camera (bright field). Image manipulation and figure generation were performed in Fiji (RRID:SCR_002285), Adobe Photoshop (RRID: SCR_014199) and Adobe Illustrator (RRID:SCR_014198). Cell number counting, nociceptor arbor measurements, and fluorescence measurements were performed in Fiji. For section histology experiments, $n$ = 3–6 sections per animal from three animals. Central arbor height/width measurements were taken to be the relevant axes of fitted ellipses. Column graphs, pie charts and scatter plots were generated in GraphPad Prism5. Column graphs show mean ±SEM. Statistical significance was analyzed using unpaired, two-tailed Student's $t$-tests, one-way ANOVA with Tukey's multiple comparisons, linear regression, Spearman's rank correlation test, or Chi-square tests in GraphPad Prism5 (RRID:SCR_002798). For area measurements of DiI skin labeling, fluorescence intensity thresholds were set at 10 standard deviations above mean background fluorescence, and area was measured from pixel counts above threshold in Fiji. Densitometry quantification of Western blot bands was performed using Image Lab (RRID:SCR_014210). For DRG fluorescence intensity measurements, the average fluorescence of outlined cells was normalized to mean background fluorescence (*Figure 7— figure supplement 1*).

## Acknowledgements

We thank Peter Dong for illustrating the model in *Figure 8*. We also thank Greg Bashaw for reading and providing suggestions for this manuscript. This work was supported by National Institutes of Health (NIH) grant (NS083702, NS094224) and the Klingenstein-Simons Fellowship Award in the Neurosciences to W.L. and by NIH grant (NS092297) to W.O. I.A.S. was supported by NIH grant (K12GM081259) and Burroughs Wellcome Fund grant PDEP.

## Additional information

### Funding

| Funder | Grant reference number | Author |
| --- | --- | --- |
| National Institute of Neurological Disorders and Stroke | NS083702 | Wenqin Luo |
| Burroughs Wellcome Fund | PDEP | Ishmail Abdus-Saboor |
| National Institute of Neurological Disorders and Stroke | NS094224 | Wenqin Luo |
| National Institute of Neurological Disorders and Stroke | NS092297 | William Olson |

The funders had no role in study design, data collection and interpretation, or the decision to submit the work for publication.

### Author contributions

William Olson, Conceptualization, Funding acquisition, Investigation, Writing—original draft, Writing—review and editing; Ishmail Abdus-Saboor, Lian Cui, Conceptualization, Investigation, Writing—original draft; Justin Burdge, Tobias Raabe, Investigation; Minghong Ma, Conceptualization, Writing—review and editing; Wenqin Luo, Conceptualization, Supervision, Funding acquisition, Investigation, Writing—review and editing

## Author ORCIDs
William Olson, http://orcid.org/0000-0001-9893-9726
Wenqin Luo, http://orcid.org/0000-0002-2486-807X

## Ethics
Animal experimentation: All procedures were conducted according to an animal protocol (#804886) approved by Institutional Animal Care and Use Committee (IACUC) of the University of Pennsylvania and National Institutes of Health guidelines.

## Decision letter and Author response
Decision letter https://doi.org/10.7554/eLife.29507.031
Author response https://doi.org/10.7554/eLife.29507.032

# Additional files

## Supplementary files
• Transparent reporting form
DOI: https://doi.org/10.7554/eLife.29507.030

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
