## [Decision Letter]

Thank you for submitting your article "Sparse genetic tracing reveals regionally specific functional organization of mammalian nociceptors" for consideration by *eLife*. Your article has been favorably evaluated by a Senior Editor and three reviewers, one of whom is a member of our Board of Reviewing Editors. The following individuals involved in review of your submission have agreed to reveal their identity: Gregory Scherrer (Reviewer #2).

The reviewers have discussed the reviews with one another and the Reviewing Editor has drafted this decision to help you prepare a revised submission.

Summary:

Olson and colleagues use a combination of innovative and elegant mouse genetics, neuroanatomical, electrophysiological and behavioral techniques to investigate the anatomical organization and function of the primary afferent somatosensory neurons that express MrgprD. The authors notably generated a novel mouse line, MrgprD-CreERT2, which provides temporally controlled genetic access to this class of DRG neurons, and reveals their detailed anatomy through sparse labeling techniques. Consistent with previous work from this research group, the anatomical aspects of the study are beautifully performed and the results very convincing.

Essential revisions:

The authors also make a heroic effort to assess the functional correlate of the anatomical differences that they identified. Unfortunately, there is a significant weakness in these studies, which question the conclusions that are drawn. Most importantly, based on stimulation threshold differences between the paw and leg for evoking a withdrawal behavior in response to light stimulation, the authors conclude that the pattern of arborization in the dorsal horn, rather than the pattern of the peripheral arborization underlies functional differences. The authors write that: "Interestingly, our use of peripheral optogenetic stimulation likely bypasses the effects some peripheral parameters, such as the physical properties of the skin or the expression of molecular receptors, on the sensation of natural stimuli."

It is critical that in any revision the authors provide more evidence that this is the case. If light penetrates better through the paw than the thigh (to reach afferents), then the peripheral terminals may be more critical than the authors conclude. In addition, Figure 7—figure supplement 1 illustrates a very important distinction in the peripheral arborization of the afferents. The plantar paw has axons that arborize closer to the surface than the thigh. It is not even clear whether the axons reach the epidermis in thigh skin. The authors do not comment at all on this potentially critical distinction, but rather emphasize the greater density of terminal arbors in the thigh. If indeed the afferents in the thigh do not reach the superficial skin, then the light stimulation results could have another simpler explanation.

Finally, it is definitely possible that intrinsic excitability or neurotransmitter release properties differ between these two classes of MrprD+ afferents. For example, MrgprD+ afferents innervating the distal limbs could have a lower activation threshold, fire more robustly, or release glutamate more efficiently. These studies are not essential for a revision, but the issue must be considered and discussed.

---

## [Author Response]

Essential revisions:The authors also make a heroic effort to assess the functional correlate of the anatomical differences that they identified. Unfortunately, there is a significant weakness in these studies, which question the conclusions that are drawn. Most importantly, based on stimulation threshold differences between the paw and leg for evoking a withdrawal behavior in response to light stimulation, the authors conclude that the pattern of arborization in the dorsal horn, rather than the pattern of the peripheral arborization underlies functional differences. The authors write that: "Interestingly, our use of peripheral optogenetic stimulation likely bypasses the effects some peripheral parameters, such as the physical properties of the skin or the expression of molecular receptors, on the sensation of natural stimuli."It is critical that in any revision the authors provide more evidence that this is the case. If light penetrates better through the paw than the thigh (to reach afferents), then the peripheral terminals may be more critical than the authors conclude. In addition, Figure 7—figure supplement 1 illustrates a very important distinction in the peripheral arborization of the afferents. The plantar paw has axons that arborize closer to the surface than the thigh. It is not even clear whether the axons reach the epidermis in thigh skin. The authors do not comment at all on this potentially critical distinction, but rather emphasize the greater density of terminal arbors in the thigh. If indeed the afferents in the thigh do not reach the superficial skin, then the light stimulation results could have another simpler explanation.

The reviewers’ primary concern here is that we cannot conclude that the region-specific central terminal structure of these nociceptors, rather that region-specific differences in the periphery, is the major mechanism behind the increased sensitivity of the paw upon peripheral optogenetic nociceptor activation. First, it was not our intent to suggest that region-specific central terminal organization is the *only* mechanism underlying these experiments. We do believe, however, that it is likely to be an important mechanism based on our data. We have revised our interpretations of this experiment in the text to better clarify our conclusions.Further, we respectfully argue that the specific alternative explanation proposed here (that paw nociceptors terminate closer to the skin surface and are therefore more easily activated) is not the case. Paw glabrous neurites terminate much *farther* from the skin surface than hairy skin neurites, due to the much thicker outer keratinized layer in glabrous compared to hairy skin. We have thus updated the images in Figure 8—figure supplement 1 to more clearly show space between the nerve terminals and the skin surface in these two locations, and have also quantified the depth of neurite termination in these two locations.

Our overall model proposes, as taken from the Abstract, that region-specific nociceptor central terminal structure “is well correlated with a heightened signal transmission for plantar paw circuits, as revealed by both spinal cord slice recordings and behavior assays. Taken together, our results elucidate a novel somatotopic functional organization of the mammalian pain system and suggest that regional central arbor structure could facilitate the magnification of plantar paw regions to contribute to regional pain processing.” Our peripheral optogenetic experiment is presented to show that, when stimulated in the skin, the plantar paw is a region of heightened sensitivity for MrgprD^+^ afferents. Given that regional sensitivity for these circuits has not previously been assayed in mice, this provides evidence of region-specific functional differences, supporting that the anatomical difference we revealed might be functionally relevant. Given that (1) [24]our systematic anatomical analysis found major region-specific differences in the central terminals but no obvious anatomical mechanisms in the periphery (further supported by new data we have generated comparing the neuron density between these regions, see below) and (2) [25]our physiological experiments in Figure 7 demonstrate heightened transmission for plantar paw regions when only the central terminals are activated, our work can propose central terminal organization as one possible structural mechanism for region-specific pain processing. We tried to be cautious while writing this manuscript to avoid stating that there are no regional peripheral or other mechanisms at play. In any case, the presence of such mechanisms would not detract from the model we have proposed.

Finally, it is definitely possible that intrinsic excitability or neurotransmitter release properties differ between these two classes of MrprD+ afferents. For example, MrgprD+ afferents innervating the distal limbs could have a lower activation threshold, fire more robustly, or release glutamate more efficiently. These studies are not essential for a revision, but the issue must be considered and discussed.

We agree with the reviewers and have incorporated the point in the revised Discussion.As mentioned above, our model concludes that the region-specific organization we found is correlated with increased signal transmission for plantar paw circuits. We cannot rule out other kinds of physiological differences, such as proposed here, and we therefore suggest the anatomical differences we found to be one potential structural mechanism for region-specific pain processing.